# The relationship between hemoglobin and $\dot{V}O_2max$: A systematic review and meta-analysis

Kevin L. Webb[1,2], Ellen K. Gorman[1], Olaf H. Morkeberg[1], Stephen A. Klassen[3], Riley J. Regimbal[1], Chad C. Wiggins[1], Michael J. Joyner[1,2], Shane M. Hammer[4], Jonathon W. Senefeld[1,2,5]*

**1** Department of Anesthesiology and Perioperative Medicine, Mayo Clinic, Rochester, Minnesota, United States of America, **2** Department of Physiology and Biomedical Engineering, Mayo Clinic, Rochester, Minnesota, United States of America, **3** Department of Kinesiology, Brock University, St. Catharines, Ontario, Canada, **4** Department of Kinesiology, Applied Health, and Recreation, Oklahoma State University, Stillwater, Oklahoma, United States of America, **5** Department of Kinesiology and Community Health, University of Illinois at Urbana-Champaign, Urbana, Illinois

☯ These authors contributed equally to this work.
\* senefeld@illinois.edu

**Editor:** Daniel Boullosa, Universidad de León Facultad de la Ciencias de la Actividad Física y el Deporte: Universidad de Leon Facultad de la Ciencias de la Actividad Fisica y el Deporte, SPAIN

**Data Availability Statement:** All relevant data are within the paper and its Supporting Information

## Abstract

### Objective

There is widespread agreement about the key role of hemoglobin for oxygen transport. Both observational and interventional studies have examined the relationship between hemoglobin levels and maximal oxygen uptake ($\dot{V}O_2max$) in humans. However, there exists considerable variability in the scientific literature regarding the potential relationship between hemoglobin and $\dot{V}O_2max$. Thus, we aimed to provide a comprehensive analysis of the diverse literature and examine the relationship between hemoglobin levels (hemoglobin concentration and mass) and $\dot{V}O_2max$ (absolute and relative $\dot{V}O_2max$) among both observational and interventional studies.

### Methods

A systematic search was performed on December 6[th], 2021. The study procedures and reporting of findings followed Preferred Reporting Items for Systematic Reviews and Meta-Analyses (PRISMA) guidelines. Article selection and data abstraction were performed in duplicate by two independent reviewers. Primary outcomes were hemoglobin levels and $\dot{V}O_2max$ values (absolute and relative). For observational studies, meta-regression models were performed to examine the relationship between hemoglobin levels and $\dot{V}O_2max$ values. For interventional studies, meta-analysis models were performed to determine the change in $\dot{V}O_2max$ values (standard paired difference) associated with interventions designed to modify hemoglobin levels or $\dot{V}O_2max$. Meta-regression models were then performed to determine the relationship between a change in hemoglobin levels and the change in $\dot{V}O_2max$ values.

files. Additionally, the data underlying the results presented in the study are publicly available from the references listed herein.

**Funding:** This work was supported by the National Heart, Lung, and Blood Institute T-32-HL105355 (K.L.W.), R-35-HL139854 (C.C.W. and M.J.J.), and F-32-HL154320 (J.W.S.). The funders had no role in study design, data collection and analysis, decision to publish, or preparation of the manuscript.

**Competing interests:** The authors have declared that no competing interests exist.

## Results

Data from 384 studies (226 observational studies and 158 interventional studies) were examined. For observational data, there was a positive association between absolute $\dot{V}O_2max$ and hemoglobin levels (hemoglobin concentration, hemoglobin mass, and hematocrit ($P<0.001$ for all)). Prespecified subgroup analyses demonstrated no apparent sex-related differences among these relationships. For interventional data, there was a positive association between the change of absolute $\dot{V}O_2max$ (standard paired difference) and the change in hemoglobin levels (hemoglobin concentration ($P<0.0001$) and hemoglobin mass ($P = 0.006$)).

## Conclusion

These findings suggest that $\dot{V}O_2max$ values are closely associated with hemoglobin levels among both observational and interventional studies. Although our findings suggest a lack of sex differences in these relationships, there were limited studies incorporating females or stratifying results by biological sex.

## Introduction

Maximal oxygen uptake ($\dot{V}O_2max$) represents the highest rate of oxygen uptake and utilization during large muscle-mass exercise [1]. The concept of $\dot{V}O_2max$ was first described by the Nobel prize laureate A.V. Hill in the early 1920's [2, 3] and is a preeminent physiological variable in the field of human integrative physiology [4–6]. In addition to $\dot{V}O_2max$ describing cardiorespiratory fitness among athletes and otherwise healthy humans [7–10], $\dot{V}O_2max$ also serves as a powerful prognostic index of both the life span and health span among humans, including those with chronic disease [11, 12]. Although there is debate about which steps along the oxygen transport pathway limit $\dot{V}O_2max$ in healthy humans, there is widespread agreement about the key role of oxygen transport via hemoglobin in the blood [13, 14]. In this context, the relationship between hemoglobin levels and $\dot{V}O_2max$ in humans has been extensively studied [15–17].

### Rationale

Many different approaches have been used to study the relationship between hemoglobin levels and $\dot{V}O_2max$ in humans, including both observational studies and interventional studies. When designing a study to estimate the causal effect of an intervention on a physiological outcome (for example, the change in hemoglobin levels on $\dot{V}O_2max$), interventional studies are generally considered to be the least susceptible to bias [18]. For interventional studies, researchers control the assignment of the treatment or exposure which may limit bias associated with unmeasured confounders. Observational study designs, however, are more susceptible to unmeasured confounding influences, and generally do not involve a researcher-controlled intervention. Thus, interventional and observational study designs are inherently different from an epistemological standpoint [19]. Although a strong relationship between hemoglobin levels and $\dot{V}O_2max$ in humans has been demonstrated in both observational studies [16, 20, 21] and interventional studies [22–24], some uncertainty remains. As an example,

although it is clear that an increase in hemoglobin levels is associated with an increase in $\dot{V}O_2$max (a process referred to as 'blood doping' in sport [24]), uncertainty remains about the potential relationship between hemoglobin levels and $\dot{V}O_2$max among other interventions (e.g., dietary supplementation). In this framework, there are ambiguous areas associated with biological diversity and heterogenous study designs, and an unbiased synthesis of information may provide key insights from the scientific corpus [25]. Thus, the present work was performed to provide a comprehensive systematic review and meta-analysis on the relationship between hemoglobin levels and $\dot{V}O_2$max in humans.

### Objectives

This systematic review and meta-analysis aimed to evaluate the relationship between hemoglobin levels and $\dot{V}O_2$max by pooling data from observational studies and interventional studies. Moreover, as a secondary objective, prespecified analyses aimed to determine whether this relationship is influenced by population characteristics (i.e., biological sex) or intervention characteristics (subsequently defined in methods section) which may modify hemoglobin levels or $\dot{V}O_2$max.

## Methods

### Registration and protocol

This systematic review and meta-analysis followed the recommendations of and reported findings according to the Preferred Reporting Items for Systematic Reviews and Meta-analyses (PRISMA) guidelines [26]. The study protocol has been registered in the International Prospective Register of Systematic Reviews, 'PROSPERO', (CRD42022329010). All changes to the study protocol are reported in the Methods section. In accordance with the Code of Federal Regulations, 45 CFR 46.102, this study was exempt from obtaining institutional review board approval from Mayo Clinic and from obtaining informed patient consent because it represents secondary use of publicly available data sets.

### Information sources and search strategy

On December 6, 2021, MEDLINE was searched for eligible articles by three authors (S.A.K., C. C.W., J.W.S.). Keywords used in the search included: [(oxygen consumption) OR (oxygen uptake) OR ($VO_2$) OR ($VO_2$max)] AND [(hemoglobin) OR (hematocrit) OR (hemodilution) OR (transfusion) OR (venesection) OR (anemia) OR (polycythemia) OR (erythrocythemia) OR (erythropoietin) OR (blood loss) OR (blood doping)] AND (exercise). Only published material was identified and exported to a web-based systematic review management software (Covidence, Melbourne, Australia). The search was not limited by study design or publication period. To be eligible for inclusion, full-text translations must have been available in English. References of included articles were examined for potential inclusion.

### Eligibility criteria

Eligible participants included healthy adults (age 18 years or older) and adults with disordered erythropoiesis (anemia, polycythemia, iron deficiency with anemia, iron deficiency without anemia, or thalassemia) without secondary pathologies.

Eligible articles met prespecified inclusion criteria. Eligible articles reported hemoglobin levels (hemoglobin concentration or hemoglobin mass) and $\dot{V}O_2$max achieved during exercise. Eligible $\dot{V}O_2$max tests typified a standard $\dot{V}O_2$max protocol and included a graded or

incremental exercise test to task failure with continuous pulmonary gas exchange measurements. Task failure criteria included a combination of at least two of the following criteria: 1) "classical" plateau in $\dot{V}O_2$; 2) elevated respiratory exchange ratio (RER, greater than 1.0, 1.1, or 1.15); 3) near maximal, self-reported rating of perceived effort (e.g., Borg Scale$_{6-20}$ $\geq$ 17); and 4) maximum heart rate greater than 90% of age-predicted maximum heart rate. In this framework, studies that assessed $\dot{V}O_2$peak or *estimated* $\dot{V}O_2$max were not eligible for inclusion.

Eligible exercise modalities which may be associated with an accurate representation of $\dot{V}O_2$max included running, cycling, swimming, rowing, inclined walking, and exercise using a large muscle-mass ergometer (ski, kayak, or row ergometer). Eligible $\dot{V}O_2$max tests were performed in standard environmental conditions. Thus, ineligible $\dot{V}O_2$max tests included tests performed using non-normoxic inspirates, an altitude greater than approximately sea level, or substantial deviation from room temperature (~20°C).

Eligible hemoglobin concentrations were determined using blood samples obtained via venipuncture or arterial catheterization. Measurements of hemoglobin concentration are associated with both good stability and validity [27, 28], and thus, are often considered the gold-standard measurement of circulating hemoglobin. Notably, however, there is heterogeneity in methodological assessments and associated biological variability due to factors such as the source of blood sample, body positioning during blood collection, circadian variation, and measurement device [29]. Eligible hemoglobin mass values were determined using either a carbon monoxide rebreathing protocol [30] or dye dilution with a fluorescent or radioactive tracer [31]. Similar to measurements of hemoglobin concentration; however, these techniques are subject to considerable heterogeneity in methodological assessment. Additionally, confounding physiological factors such as changes in circulating volumes and incomplete distribution/mixing of the tracer may negatively influence the reliability of these techniques [32].

## Selection process and data collection process

Both the selection process and data abstraction process were performed in duplicate and independently by two reviewers from a cohort of seven potential reviewers (K.L.W., E.K.G., O.H. M, S.A.K., R.J.R, S.M.H, and J.W.S.). Conflicts regarding article selection and data abstraction were independently verified by a third reviewer. Disagreements were discussed until consensus. Further information on the article selection process is presented in Fig 1.

## Data items

Data abstraction was performed using a standardized data abstraction form. Abstracted data included participant characteristics (sample size, sex, age, height, weight, body mass index, absolute $\dot{V}O_2$max, and relative $\dot{V}O_2$max) and hematological characteristics (hemoglobin levels (both hemoglobin concentration and hemoglobin mass), and hematocrit) as available.

If an article reported multiple measurements of $\dot{V}O_2$max for the same condition associated with a single measurement of hemoglobin level, the greatest value of $\dot{V}O_2$max was used for analyses. If an article reported hematological data from both arterial and venous blood samples, data associated with the arterial blood sample(s) was used for analyses.

If data associated with primary outcomes (hemoglobin levels and $\dot{V}O_2$max values) were not numerically reported in the text, authors were contacted via email to obtain numerical data. If the data could not be provided but were presented in figures, WebPlotDigitizer [33] was used to extract means and standard deviations. If data were reported with standard error (SE), the following equation was used to convert data to standard deviation (SD): $SD = SE \times \sqrt{n}$.

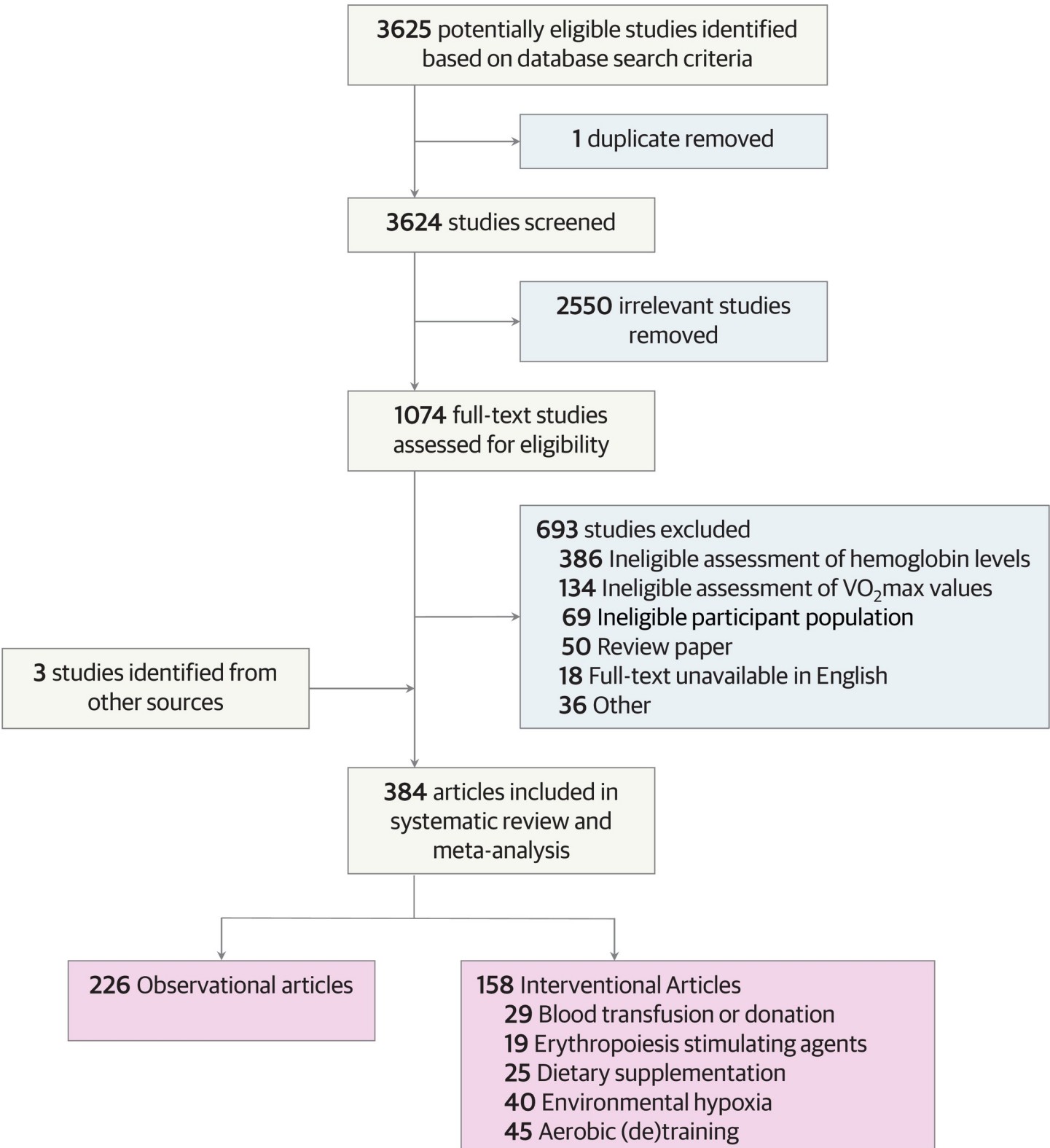

**Fig 1. PRISMA flow diagram.** Flow diagram displaying the selection of articles through different phases of the systematic review, and categorization of included articles.

**Article categorization.** Included articles were categorized as observational or interventional according to study design [18]. Observational articles included studies which reported the primary outcomes (hemoglobin levels and $\dot{V}O_2$max values) without respect to a defined event or intervention. Interventional articles included allocation to a clearly defined event or intervention and reported the primary outcomes (hemoglobin levels and $\dot{V}O_2$max values) before and after the clearly defined event or intervention. Interventional articles were further categorized into five mutually exclusive subgroups according to the intervention: 1) blood transfusion or donation, 2) erythropoiesis stimulating agents, 3) dietary supplementation, 4) environmental hypoxia, or 5) aerobic (de)training.

## Study risk of bias assessment and reporting bias assessment

Risk of bias assessment was conducted using the Risk Of Bias In Non-Randomized Studies—of Interventions (ROBINS-I) [34] for interventional studies and the Risk Of Bias In Non-randomized Studies–of Exposures (ROBINS-E) [35] for observational studies. Two reviewers from a cohort of three potential reviewers (K.L.W., E.K.G, J.W.S.) applied the risk of bias assessment independently, and data were independently verified by a third reviewer. Disagreements were discussed until consensus. Egger's [regression] test was used to assess potential publication bias in meta-analyses.

## Effect measures

Applied meta-regressions used absolute and relative $\dot{V}O_2$max as the effect measure. Meta-analysis models used the standard paired difference as the effect measure to quantify changes in $\dot{V}O_2$max following a performed intervention. The tau-squared ($\tau^2$) statistic was used to estimate between-study variance and the Higgins I-squared statistic ($I^2$) was used to estimate *relative* heterogeneity within meta-analyses [36].

## Synthesis methods

For the primary outcomes (hemoglobin levels and $\dot{V}O_2$max values), we performed meta-analysis and meta-regression using random-effects models to account for variability between studies [37]. Analyses were performed separately for observational studies and interventional studies. Outcome data were pooled from each study to provide one single data point. A minimum of two studies were required to perform meta-analyses [38]. Additional prespecified, exploratory analyses were performed stratified by biological sex by pooling data for male and female subgroups within each study as available. Analyses were performed using Comprehensive Meta-Analysis software (CMA 3.3, Biostat, Englewood, New Jersey, USA). Figures were created using SigmaPlot software (SigmaPlot 12.5, Systat Software Inc, Chicago, Illinois, USA). Results are reported with 95% confidence intervals (CIs), statistical significance was set at α = 0.05, and all tests were two-tailed.

**Observational studies.** For analyses of observational studies, an amalgam was created using all observational articles and data associated with interventional articles that were measured before the defined event or intervention (so called "baseline data"). The primary analysis investigated the relationship between hemoglobin levels and $\dot{V}O_2$max values. Because hemoglobin level was described by two metrics (hemoglobin concentration and hemoglobin mass) and $\dot{V}O_2$max was described by two metrics (absolute $\dot{V}O_2$max and relative $\dot{V}O_2$max), the primary analysis was associated with four separate comparisons. Meta-regression models were performed to determine heterogeneity of $\dot{V}O_2$max values associated with hemoglobin levels. Exploratory meta-regression models were then performed with subgroups stratified by

biological sex. Results of meta-regression models are presented as (slope coefficient; 95% Confidence Interval (CI); $R^2$; $\tau^2$; $P$-value; $n$ (number of studies or subgroups)).

**Interventional studies.** For analyses of interventional studies, standard paired difference was used to quantify the change of absolute $\dot{V}O_2$max and relative $\dot{V}O_2$max (separately) in response to the defined, researcher-controlled event or intervention. Meta-analyses were performed for subgroups of interventional studies as defined in the '*Article Categorization*' section herein. Results of meta-analyses are presented as (pooled standard paired difference; 95% CI; $Z$-value; $\tau^2$; $P$-value; $n$ (number of studies)).

Meta-regression models were performed to assess the relationship between study-level covariates (hemoglobin levels) and the effect size of the change in $\dot{V}O_2$max values (standard paired difference). Similar to analyses of observational studies, the primary meta-regression models were associated with four separate comparisons because both hemoglobin levels and $\dot{V}O_2$max values are described by two separate variables.

## Results

### Study selection and study characteristics

The process of study selection and article categorization is represented in the PRISMA flow diagram (Fig 1). A total of 3,625 articles were identified in the initial literature search, and duplicates were removed ($n$ = 1). After screening titles and abstracts, 2,550 articles were deemed ineligible for inclusion. Of the remaining 1,074 full text articles, 381 articles were included. The references of the included 381 articles were then carefully inspected, and an additional 3 articles were deemed eligible for inclusion. Thus, 384 articles were included. Of the 384 included articles, 226 articles were categorized as observational [15, 20, 39–262], and 158 articles were categorized as interventional. The 158 interventional articles were further categorized into five mutually exclusive subgroups according to the intervention: 1) blood transfusion or donation ($n$ = 29) [263–291], 2) erythropoiesis stimulating agents ($n$ = 19) [292–310], 3) dietary supplementation ($n$ = 25) [311–335], 4) environmental hypoxia ($n$ = 40) [336–375], or 5) aerobic (de)training ($n$ = 45) [376–420].

Ten of the included articles reported data for participants with disordered erythropoiesis ($n$ = 6 beta-thalassemia, $n$ = 2 iron deficiency anemia, $n$ = 2 iron deficiency non-anemia). Within these ten articles, no measurements of hemoglobin mass were reported.

### Risk of bias in studies

The results of the risk of bias assessment for observational articles and interventional articles are presented in S1 and S2 Tables, respectively. Among observational articles, 83% (188 of 226 articles) had a low risk of bias, 12% (28 of 226 articles) had a moderate risk of bias, and 4% (10 of 226 articles) had a high risk of bias. The moderate and high risk of bias were primarily associated with selection of participants and selection of reported results, primarily owing to the inclusion of participants with disordered erythropoiesis. Among interventional articles, 86% (136 of 158 articles) had a low risk of bias, 13% (20 of 158 articles) had a moderate risk of bias, and 1% (2 of 158 articles) had a high risk of bias. The moderate and high risk of bias were primarily associated with missing data and selection of reported results.

Egger's regression tests demonstrated no significant publication bias among any interventional subgroups, including: 1) blood transfusion or donation (intercept, -0.805; $P$ = 0.558; $n$ = 29 studies), 2) erythropoiesis stimulating agents (intercept, 1.573; $P$ = 0.136; $n$ = 19 studies), 3) dietary supplementation (intercept, -1.070; $P$ = 0.394; $n$ = 25 studies), 4) environmental

hypoxia (intercept, 0.290; $P$ = 0.768; $n$ = 40 studies), and 5) aerobic (de)training (intercept, 0.307; $P$ = 0.741; $n$ = 45 studies).

## Results of syntheses

**Observational analyses.** For observational analyses, data were amalgamated from 226 observational articles and "baseline data" from 158 interventional articles. Observational data included 5,990 male participants and 3,310 female participants. Of the 384 included articles, 228 articles reported only data associated with male participants, 52 articles reported only data associated with female participants, 78 articles reported pooled data for males and females, and 26 articles reported data stratified by biological sex.

Meta-regression models demonstrated a strong, positive association between study-level data of $\dot{V}O_2$max values and hemoglobin levels (hemoglobin mass (Fig 2) and hemoglobin concentration (S1 Fig)), see Table 1. For data obtained from participants with disordered erythropoiesis, meta-regression models demonstrated a positive association between study-level data of hemoglobin concentration and absolute $\dot{V}O_2$max (slope coefficient, 0.255; 95% CI, 0.018 to 0.491; $R^2$ = 0.42; $\tau^2$ = 0.28; $P$ = 0.035; $n$ = 6 studies). When excluding participants with disordered erythropoiesis, a positive association between hemoglobin concentration and absolute $\dot{V}O_2$max remained (slope coefficient, 0.402; 95% CI, 0.273 to 0.532; $R^2$ = 0.28; $\tau^2$ = 0.79; $P$<0.0001; $n$ = 193 studies).

Meta-regression models also demonstrated a positive association between study-level data of hematocrit and both absolute $\dot{V}O_2$max (slope coefficient, 0.146; 95% CI, 0.091 to 0.200; $R^2$ = 0.19; $\tau^2$ = 1.14; $P$<0.0001; $n$ = 136 studies) and relative $\dot{V}O_2$max (slope coefficient, 1.449; 95% CI, 0.965 to 1.932; $R^2$ = 0.20; $\tau^2$ = 105.69; $P$<0.0001; $n$ = 195 studies; S2 Fig). Exploratory analyses stratifying meta-regression models by biological sex found no sex differences in the relationship between hemoglobin mass and $\dot{V}O_2$max values (Table 1).

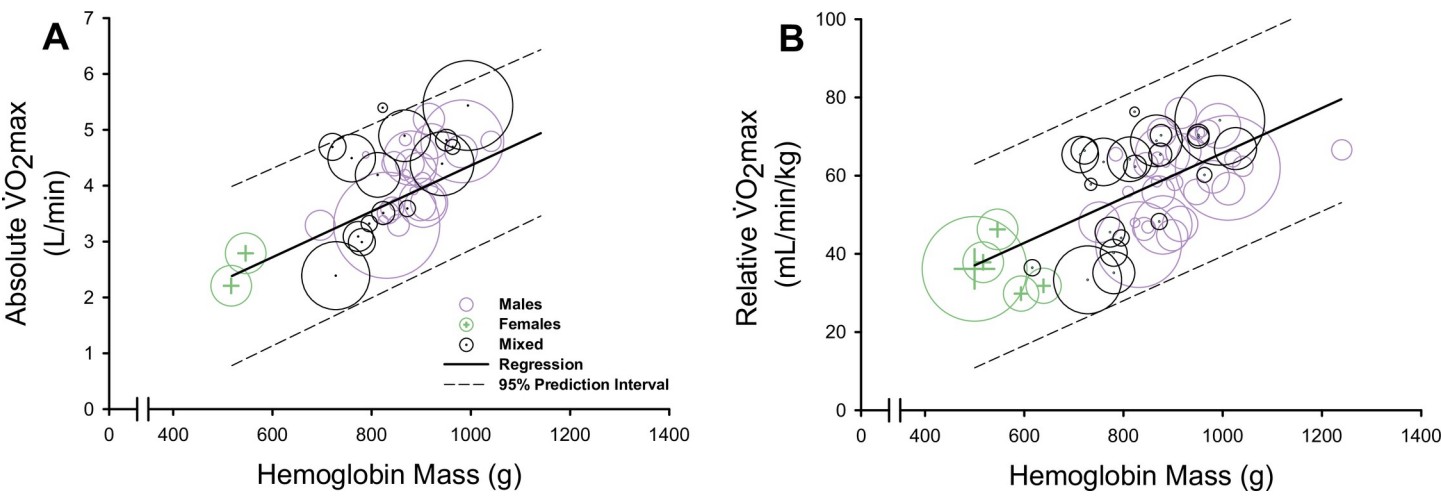

**Fig 2. The association between hemoglobin mass and maximal oxygen uptake** ($\dot{V}O_2$max). Bubble plots and meta-regressions displaying the positive association between hemoglobin mass and both absolute $\dot{V}O_2$max (A) and relative $\dot{V}O_2$max (B). Data for males are represented as purple bubbles, data for females are represented as green bubbles with plus sign symbols, and data for studies presenting males and females pooled (mixed) are represented as black bubbles with middle dot symbols. Each bubble represents a group from a single study and the size of bubbles represents the number of participants within the group. The solid line indicates the meta-regression line, and the dashed lines indicate the 95% prediction interval associated with the meta-regression.

**Table 1. Hemoglobin levels as covariates of absolute and relative maximal oxygen uptake ($\dot{V}O_2$max) derived from observational analyses.**

| | Absolute $\dot{V}O_2$max (L·min$^{-1}$) | | | | |
|---|---|---|---|---|---|
| | Coefficient [95% CI] | P value | $R^2$ | $\tau^2$ | n |
| **Hemoglobin mass** | | | | | |
| All data | 0.005 [0.003, 0.007] | <0.0001 | 0.49 | 0.34 | 42 |
| Males | 0.006 [0.004, 0.009] | <0.0001 | 0.66 | 0.41 | 23 |
| Females | 0.011 [0.008, 0.015] | <0.0001 | 0.92 | 0.31 | 8 |
| **Hemoglobin concentration** | | | | | |
| All data | 0.404 [0.298, 0.510] | <0.0001 | 0.29 | 0.77 | 199 |
| Males | 0.148 [0.022, 0.274] | 0.022 | 0.15 | 0.51 | 141 |
| Females | 0.226 [0.060, 0.393] | 0.008 | 0.22 | 0.31 | 41 |
| | Relative $\dot{V}O_2$max (mL·min$^{-1}$·kg$^{-1}$) | | | | |
| **Hemoglobin mass** | | | | | |
| All data | 0.058 [0.039, 0.077] | <0.0001 | 0.55 | 130.27 | 58 |
| Males | 0.054 [0.024, 0.084] | <0.0001 | 0.19 | 67.75 | 37 |
| Females | 0.137 [0.016, 0.258] | 0.027 | 0.34 | 28.44 | 8 |
| **Hemoglobin concentration** | | | | | |
| All data | 4.649 [2.650, 6.648] | <0.0001 | 0.10 | 121.87 | 284 |
| Males | 0.882 [-1.120, 2.884] | 0.388 | 0.02 | 83.77 | 175 |
| Females | 5.999 [3.249, 8.728] | <0.0001 | 0.45 | 67.59 | 63 |

Data depict results of independent meta-regressions in determination of hemoglobin levels (hemoglobin mass and hemoglobin concentration) as covariates of absolute $\dot{V}O_2$max and relative $\dot{V}O_2$max. Data were pooled from each study to provide one single data point for 'All data' derived coefficients (n denotes the number of studies). Additional independent analyses were performed for subgroups of data reporting males only and females (n denotes the number of subgroups). Abbreviations: CI, confidence interval.

**Interventional analyses.** For analyses of interventional articles, data from 158 articles were included. From pooled interventional data, the meta-analysis model demonstrated a significant increase in absolute $\dot{V}O_2$max following the performed interventions (pooled standard paired difference, 0.183; 95% CI, 0.056 to 0.310; $Z = 2.829$; $\tau^2 = 0.24$; $I^2 = 11\%$; $P = 0.005$; $n = 106$ studies). The meta-regression model demonstrated a positive association between the change in hemoglobin concentration and the effect size of the change in absolute $\dot{V}O_2$max (standard paired difference), (slope coefficient, 0.271; 95% CI, 0.158 to 0.385; $R^2 = 0.25$; $\tau^2 = 0.19$; $P<0.0001$; $n = 100$ studies). An additional meta-regression model demonstrated a positive association between the change in hemoglobin mass and the effect size of the change in absolute $\dot{V}O_2$max (standard paired difference), (slope coefficient, 0.008; 95% CI, 0.002 to 0.014; $R^2 = 0.04$; $\tau^2 = 0.31$; $P = 0.006$; $n = 24$ studies). The 158 interventional articles were further categorized into five mutually exclusive subgroups according to the intervention. Thus, prespecified subanalyses were performed to examine the relationship between changes in hemoglobin levels and changes in $\dot{V}O_2$max values within the interventional subgroups, independently.

**Blood transfusion or donation.** Analyses included 29 articles associated with blood transfusion or donation. Of the 29 articles, 7 articles examined blood transfusion, 17 articles examined blood donation, and 5 articles examined both blood transfusion and donation. Overall, data were reported for a total of 310 male participants and 84 female participants. Only one article reported data for females only, limiting potential analyses stratified by sex.

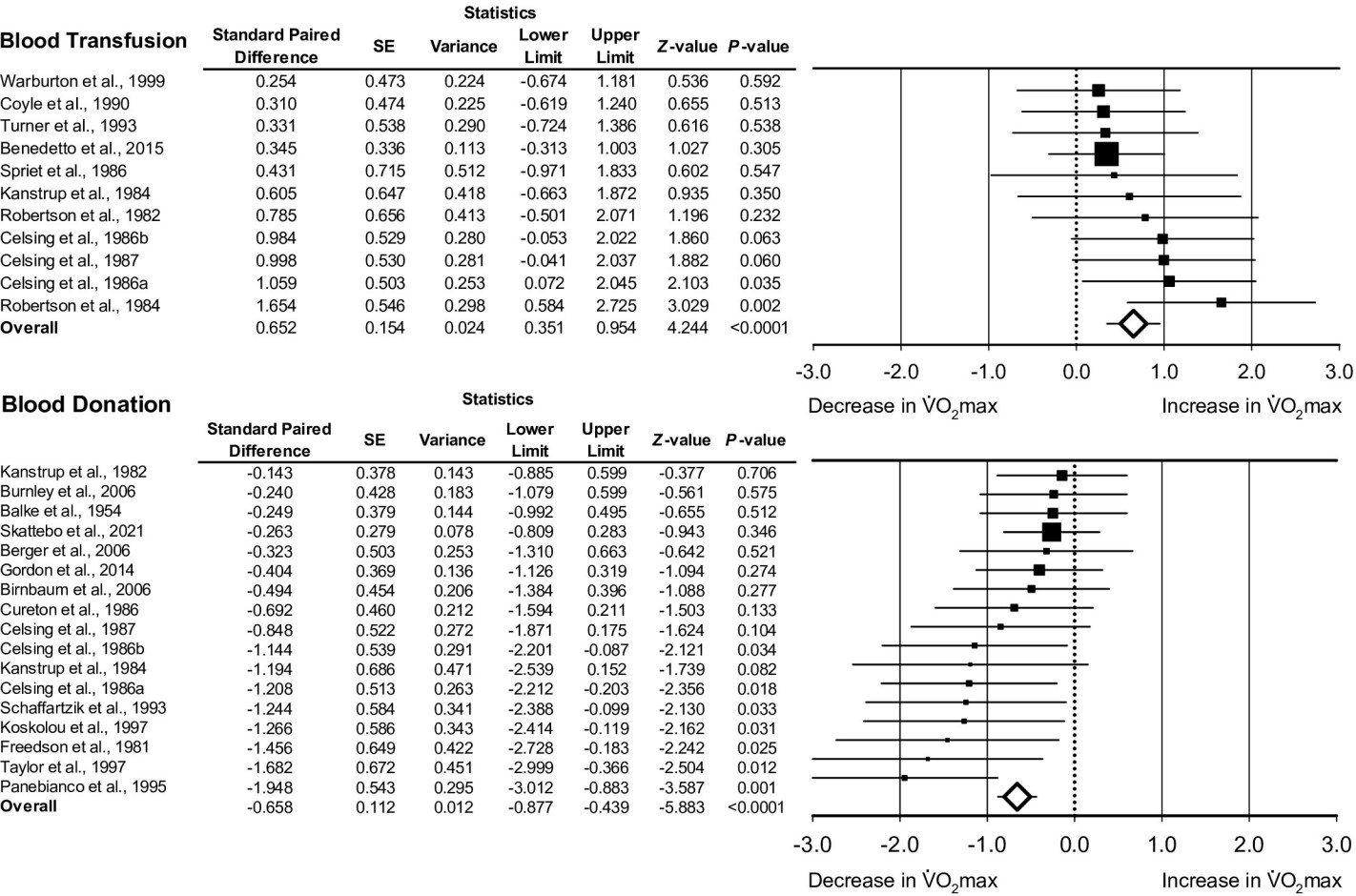

**Fig 3. The association between blood donation or transfusion and change in maximal oxygen uptake (absolute V̇O₂max).** Forest plots depicting the effect size of the change of absolute V̇O₂max (standard paired difference) following blood transfusion or donation. Different size symbols indicate relative weights used in meta-analyses and are proportional to study size. Abbreviations: SE, standard error.

For interventional articles associated with blood transfusion, the median amount of blood transfused was 500 mL (range, 400 to 900 mL). Data for the meta-analysis of blood transfusion studies were associated with an increase in absolute V̇O₂max (pooled standard paired difference, 0.652; 95% CI, 0.351 to 0.954; $Z = 4.244$; $\tau^2 = 0.02$; $I^2 = 2\%$; $P<0.0001$; $n = 11$ studies; Fig 3). When removing one study examining blood transfusion among a clinical population (participants with beta-thalassemia), a significant increase in absolute V̇O₂max following transfusion remained (pooled standard paired difference, 0.734; 95% CI, 0.395 to 1.073; $Z = 4.245$; $\tau^2 = 0.03$; $I^2 = 3\%$; $P<0.0001$; $n = 10$ studies).

For articles associated with blood donation, the median amount of blood drawn from the donor was 500 mL (range, 300 to 1350 mL). Data for the meta-analysis of blood donation studies were associated with a decrease in absolute V̇O₂max (pooled standard paired difference, -0.658; 95% CI, -0.877 to -0.439; $Z = -5.883$; $\tau^2 = 0.07$; $I^2 = 39\%$; $P<0.0001$; $n = 17$ studies; Fig 3).

As described above, the 29 articles included in this section were associated with 22 datasets examining blood transfusion and 12 datasets examining blood donation (34 subgroups). The

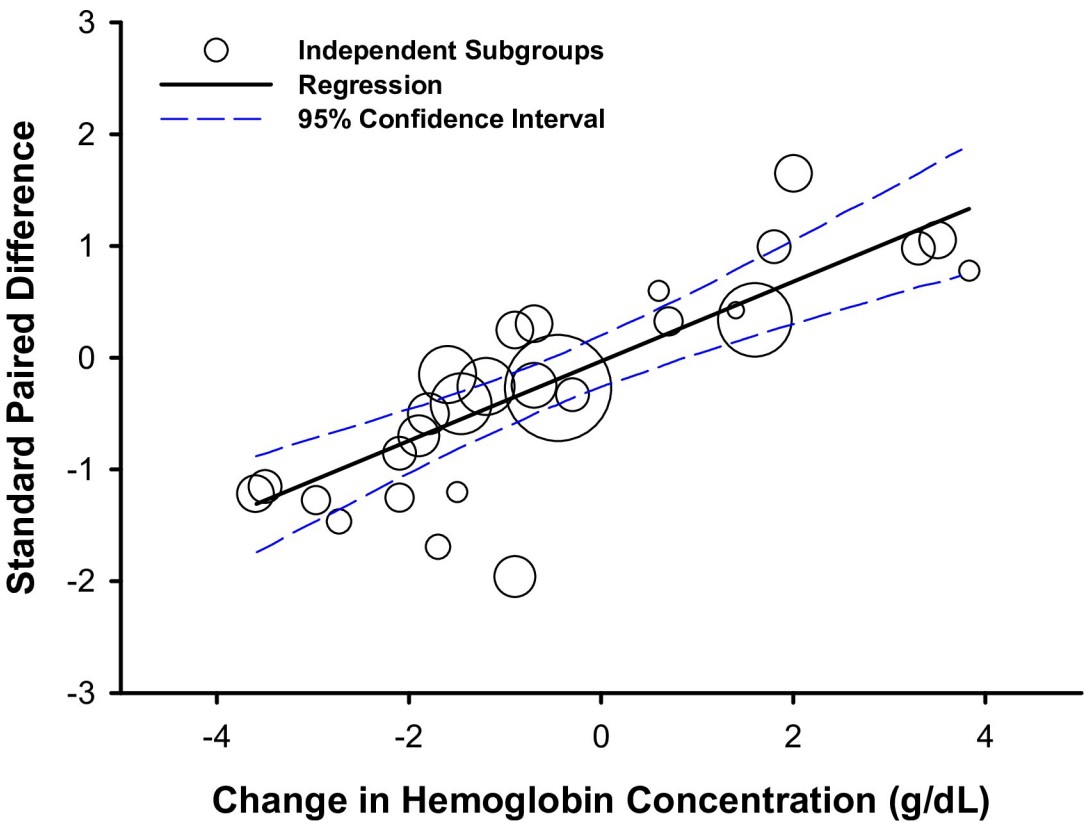

**Fig 4. The association between change in hemoglobin concentration and change in absolute maximal oxygen uptake ($\dot{V}O_2$max) following blood donation or transfusion.** Meta-regression and bubble plot depicting the relationship between the change in hemoglobin concentration and the effect size of the change in absolute $\dot{V}O_2$max (standard paired difference) among independent subgroups following blood transfusion or donation. Different size symbols indicate relative weights used in the meta-regression and are proportional to study sample size.

overall meta-regression model, including both blood transfusion and blood donation, demonstrated a positive association between the change in hemoglobin concentration and the effect size of the change in absolute $\dot{V}O_2$max (standard paired difference), (slope coefficient, 0.357; 95% CI, 0.260 to 0.454; $R^2$ = 0.99; $\tau^2$<0.01; $P$<0.0001; $n = 28$ subgroups; Fig 4A).

**Erythropoiesis stimulating agents.** Analyses included 19 articles associated with erythropoiesis stimulating agents, 18 articles examined erythropoietin supplementation and one article examined xenon inhalation. Of the 19 articles, data were reported for a total of 230 male participants and 17 female participants. No articles reported data for females only, limiting potential analyses stratified by sex. The meta-analysis model demonstrated an increase in absolute $\dot{V}O_2$max following administration of erythropoieses stimulating agents (pooled standard paired difference, 0.740; 95% CI, 0.510 to 0.969; $Z$ = 6.315; $\tau^2$ = 0.03; $I^2$ = 1%; $P$<0.0001; $n = 12$ studies; Fig 5). For studies examining the effects of erythropoietin alone, there was a significant increase in absolute $\dot{V}O_2$max following treatment (pooled standard paired difference, 0.789; 95% CI, 0.526 to 1.052; $Z$ = 5.873; $\tau^2$ = 0.03; $I^2$ = 1%; $P$<0.0001; $n = 11$ studies). Following treatment with erythropoiesis stimulating agents, the relationships between the change in hemoglobin concentration and changes in $\dot{V}O_2$max values were not statistically significant (absolute $\dot{V}O_2$max, $P$ = 0.150; relative $\dot{V}O_2$max, $P$ = 0.050).

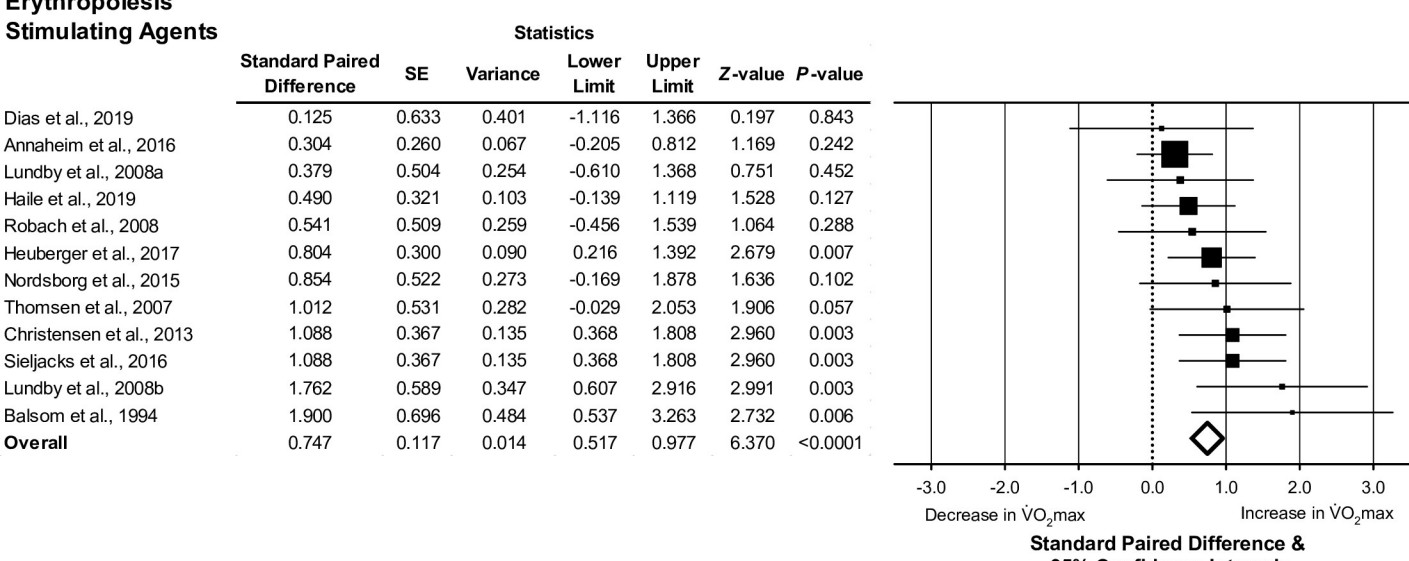

| Erythropoiesis Stimulating Agents | | | | Statistics | | | | |
|---|---|---|---|---|---|---|---|---|
| | Standard Paired Difference | SE | Variance | Lower Limit | Upper Limit | Z-value | P-value | |
| Dias et al., 2019 | 0.125 | 0.633 | 0.401 | -1.116 | 1.366 | 0.197 | 0.843 | |
| Annaheim et al., 2016 | 0.304 | 0.260 | 0.067 | -0.205 | 0.812 | 1.169 | 0.242 | |
| Lundby et al., 2008a | 0.379 | 0.504 | 0.254 | -0.610 | 1.368 | 0.751 | 0.452 | |
| Haile et al., 2019 | 0.490 | 0.321 | 0.103 | -0.139 | 1.119 | 1.528 | 0.127 | |
| Robach et al., 2008 | 0.541 | 0.509 | 0.259 | -0.456 | 1.539 | 1.064 | 0.288 | |
| Heuberger et al., 2017 | 0.804 | 0.300 | 0.090 | 0.216 | 1.392 | 2.679 | 0.007 | |
| Nordsborg et al., 2015 | 0.854 | 0.522 | 0.273 | -0.169 | 1.878 | 1.636 | 0.102 | |
| Thomsen et al., 2007 | 1.012 | 0.531 | 0.282 | -0.029 | 2.053 | 1.906 | 0.057 | |
| Christensen et al., 2013 | 1.088 | 0.367 | 0.135 | 0.368 | 1.808 | 2.960 | 0.003 | |
| Sieljacks et al., 2016 | 1.088 | 0.367 | 0.135 | 0.368 | 1.808 | 2.960 | 0.003 | |
| Lundby et al., 2008b | 1.762 | 0.589 | 0.347 | 0.607 | 2.916 | 2.991 | 0.003 | |
| Balsom et al., 1994 | 1.900 | 0.696 | 0.484 | 0.537 | 3.263 | 2.732 | 0.006 | |
| **Overall** | 0.747 | 0.117 | 0.014 | 0.517 | 0.977 | 6.370 | <0.0001 | |

**Fig 5. The association between erythropoiesis stimulating agents and change in maximal oxygen uptake (absolute V̇O₂max).** Forest plot depicting the effect size of the change in absolute V̇O₂max (standard paired difference) following administration of erythropoiesis stimulating agents. Different size symbols indicate relative weights used in meta-analyses and are proportional to study size. Abbreviations: SD, standard error.

## Dietary supplementation

Analyses included 25 articles associated with dietary supplementation—15 articles examined iron supplementation, 4 articles examined plant-based supplementation (echinacea ($n = 2$), beetroot ($n = 1$), and higenamine ($n = 1$)), 3 articles examined dietary restrictions (ketogenic diet ($n = 1$), caloric deficit ($n = 1$), and time restricted feeding ($n = 1$)), and 3 articles examined other dietary supplements (sodium phosphate ($n = 1$), vitamin C ($n = 1$), and beta-methylbutyrate ($n = 1$)). Of the 25 articles, data were reported for a total of 190 male participants and 305 female participants. Six articles reported data for males only, 13 articles reported data for females only, 4 articles reported pooled data for males and females, and 2 articles reported data stratified by biological sex. Due to an insufficient number of studies, stratified analyses were only performed to separately examine the effects of iron and echinacea supplementation.

There was a broad spectrum of heterogenous substances used for dietary supplementation, as described above. Thus, the change in hemoglobin concentration associated with dietary supplementation was heterogenous and ranged from a decrease of 1.2 g/dL to an increase of 1.3 g/dL (median, increase of 0.1 g/dL). The meta-analysis model demonstrated an increase in absolute V̇O₂max following dietary supplementation (pooled standard paired difference, 0.274; 95% CI, 0.084 to 0.464; $Z = 2.827$; $\tau^2 = 0.89$; $I^2 = 69\%$; $P = 0.005$; $n = 16$ studies). Additionally, there was an increase in absolute V̇O₂max following iron supplementation (pooled standard paired difference, 0.341; 95% CI, 0.050 to 0.631; $Z = 2.301$; $\tau^2 = 1.37$; $I^2 = 31\%$; $P = 0.021$; $n = 9$ studies). Among the two studies examining the effects of echinacea supplementation, data regarding absolute V̇O₂max were not reported. There was no significant change in relative V̇O₂max following echinacea supplementation (pooled standard paired difference, 0.330; 95% CI, -0.118 to 0.778; $Z = 1.444$; $\tau^2 = 0.29$; $I^2 = 74\%$; $P = 0.149$; $n = 2$ studies).

Meta-regressional analysis found no significant relationship between the effect size of the change in absolute V̇O₂max (standard paired difference) and the change in hemoglobin concentration ($P = 0.200$).

**Environmental hypoxia.** Analyses included 40 articles associated with environmental hypoxia, 29 articles examined high-altitude acclimatization or sojourn and 11 articles examined chronic normobaric hypoxia. Of the 40 articles, data were reported for a total of 454 male participants and 80 female participants. Only one article reported data for females only, limiting potential analyses stratified by sex. Of the 40 included articles, 13 articles reported a change in hemoglobin mass. The median change in hemoglobin mass was 20 g increase from baseline, and there was a broad range across studies (16 g decrease to 55 g increase).

The meta-analysis model demonstrated an increase in absolute $\dot{V}O_2max$ associated with environmental hypoxia (pooled standard difference, 0.203; 95% CI, 0.027 to 0.379; $Z = 2.263$; $\tau^2 = 0.20$; $I^2 = 5\%$; $P = 0.024$; $n = 24$ studies). Of the articles examining high-altitude sojourn or acclimatization, there was no significant change in absolute $\dot{V}O_2max$ (pooled standard difference, 0.004; 95% CI, -0.217 to 0.224; $Z = 0.031$; $\tau^2 = 0.13$; $I^2 = 4\%$; $P = 0.975$; $n = 16$ studies). For articles examining the effects of chronic normobaric hypoxia, there was a significant increase in absolute $\dot{V}O_2max$ (pooled standard difference, 0.550; 95% CI, 0.259 to 0.841; $Z = 3.702$; $\tau^2 = 0.01$; $I^2 = 2\%$; $P<0.0001$; $n = 8$ studies). Meta-regressional analysis demonstrated no significant relationship between the effect size of the change in $\dot{V}O_2max$ (standard paired difference) and the change in hemoglobin concentration ($P = 0.854$) or the change in hemoglobin mass ($P = 0.531$) after environmental hypoxia.

**Aerobic (de)training.** Analyses included 45 articles associated with aerobic (de)training, 36 articles examined aerobic training and 9 articles examined aerobic detraining. Of the 45 articles, data were reported for a total of 495 male participants and 167 female participants, and 9 articles reported data for females alone. The median duration of aerobic training was 5.5 weeks (range, 1 to 22 weeks) and the median duration of detraining was 6 weeks (range, 1 to 52 weeks).

The meta-analysis model demonstrated an increase in absolute $\dot{V}O_2max$ following aerobic training (pooled standard paired difference, 0.544; 95% CI, 0.378 to 0.709; $Z = 6.452$; $\tau^2 = 0.24$; $I^2 = 41\%$; $P<0.0001$; $n = 23$ studies). A separate meta-analysis model demonstrated a decrease in absolute $\dot{V}O_2max$ following aerobic detraining (pooled standard paired difference, -0.800; 95% CI, -1.088 to -0.511; $Z = -5.431$; $\tau^2 = 0.04$; $I^2 = 2\%$; $P<0.0001$; $n = 7$ studies). Pooling data from both aerobic training and detraining, there was no relationship between the change in hemoglobin concentration and the effect size of the change in absolute $\dot{V}O_2max$ ($P = 0.255$).

## Discussion

This systematic review and meta-analysis found that maximal oxygen uptake ($\dot{V}O_2max$) was positively associated with hemoglobin levels (both hemoglobin mass and hemoglobin concentration). This strong association between $\dot{V}O_2max$ and hemoglobin was observed in both observational studies and interventional studies. Given that oxygen transport to the muscle depends, in part, on hemoglobin levels, the strong relationship between $\dot{V}O_2max$ and hemoglobin is not surprising [16, 22, 263]. Notably, this relationship between $\dot{V}O_2max$ and hemoglobin was observed across a broad range of studies and was not different between males and females. These findings suggest that although complex regulation is associated with oxygen transport to muscle (including several convective and diffusive stages), there is a robust relationship between $\dot{V}O_2max$ and hemoglobin due to the central role of hemoglobin in oxygen transport.

### Observational analyses

Altogether, observational analyses highlight the importance of hemoglobin levels in determination of $\dot{V}O_2max$. Analyses demonstrate a positive association between hemoglobin

levels (hemoglobin mass and hemoglobin concentration) and $\dot{V}O_2$max. In addition, there was also a positive relationship between hematocrit and $\dot{V}O_2$max. However, we must acknowledge that these analyses are generally constrained to a range of 'healthy' hematocrit values from 30% to 55%. At hematocrit values greater than 55%, there may be a plateau, or decrease in $\dot{V}O_2$max due to increased blood viscosity and added resistance to skeletal muscle and pulmonary perfusion [421, 422]. This limitation may also be apparent when examining hemoglobin mass and hemoglobin concentration in relation to $\dot{V}O_2$max. Under conditions of large values of hemoglobin mass or concentration, it is possible that other physiological factors (i.e., oxygen diffusion capacity, cardiac output, and mitochondrial capacity) would limit $\dot{V}O_2$max.

There are well-known sex differences in physiological determinants of $\dot{V}O_2$max [423–425] —on average, males have more muscular strength/power, greater cardiac output, greater lung volume, and greater hemoglobin values (both hemoglobin concentration and hemoglobin mass), and thus greater $\dot{V}O_2$max compared to females [426, 427]. Therefore, the question remains of whether the observed relationship between hemoglobin levels and $\dot{V}O_2$max is driven by differences in hemoglobin levels, rather than simply by sex differences. When performing meta-regressional analyses stratified for biological sex, there were no sex-related differences in the association between hemoglobin levels and $\dot{V}O_2$max. These results suggest that the relationship between hemoglobin levels and VO2max, particularly the derived regression coefficients, are similar between sexes.

## Interventional analyses

We sought to examine the relationship between hemoglobin levels and $\dot{V}O_2$max values following researcher-controlled interventions designed to modify hemoglobin levels or $\dot{V}O_2$max. When pooling all interventional data, our analyses demonstrated the change in absolute $\dot{V}O_2$max is positively associated with changes in hemoglobin levels, consistent with previous findings and general physiological principles [15, 22, 263]. Among studies with a direct physiological manipulation of hemoglobin levels (interventional blood transfusion or donation), a 1 g·dL$^{-1}$ change in hemoglobin concentration corresponded to a ~5% change in absolute $\dot{V}O_2$max. Notably, alterations in both blood volume and plasma volume among the blood donors and recipients likely contribute to the observed change in VO2max.

Interventional studies that did not involve a direct physiological manipulation of hemoglobin levels had more divergent findings. Overall, there was no significant relationship between the change in hemoglobin concentration and subsequent percent change in absolute $\dot{V}O_2$max within studies examining erythropoieses stimulating agents, dietary supplementation, environmental hypoxia, or aerobic (de)training. These findings may be due, in part, to potential alterations in blood and plasma volumes that may occur during these interventions [93, 263, 402]. For instance, high-altitude acclimatization is associated with considerable interindividual variability in plasma volume reduction (and blood volume reduction) [428], which often leads to a greater hemoglobin concentration without significant changes in hemoglobin mass. However, a reduction in blood volume limits maximum cardiac output, blunting improvements in $\dot{V}O_2$max [429, 430]. In a similar instance, aerobic training often improves $\dot{V}O_2$max, despite a reduction in hemoglobin concentration due to increased blood volume [431, 432]. Therefore, the association between changes in hemoglobin levels and changes in $\dot{V}O_2$max values is often contingent on alterations of other circulatory parameters such as blood volume and cardiac output during exercise.

## Limitations

This systematic review and meta-analysis has several limitations. First, our primary literature search was conducted using one database. Second, we limited the focus of our primary analyses to a single relationship between hemoglobin and $\dot{V}O_2$max, and thus did not consider the potential confounding effects of other relevant physiological factors, such as blood volume, plasma volume, maximum cardiac output, and muscle oxygen diffusivity. Alterations among these variables likely influence the relationship between hemoglobin and $\dot{V}O_2$max. In this context, our analyses were simplistic by design. Additionally, we did not include patient-level data, and this limited the scope of our analytical models. Our methodological approach enabled the inclusion of a heterogenous collection of studies that comprised diverse participant characteristics and study designs. In this framework, including a large diversity of literature may be associated with broad generalizability of these findings, but this approach also limited mechanistic insights that may be gleaned from these analyses. For example, the analyses of dietary supplementation were heterogeneous, potentially owing to between-study differences in the type of dietary supplementation, dose of dietary supplement, duration of intervention, and other putative confounding factors. Our approach did not include these analyses primarily because data were largely unavailable.

Third, the measurement of hemoglobin may be variable. There are several metrics and methods used to quantify hemoglobin which may lead to variability between studies, and the measurement of hemoglobin is associated with biological diversity due to confounding physiological factors, such as changes in circulatory volume, circadian variations, and body positioning during blood collection [29]. In this context, we believe the high level of concordance among study outcomes despite the heterogeneity associated with hemoglobin measurement offers compelling evidence for the relationship between hemoglobin levels and $\dot{V}O_2$max.

## Conclusion

This systematic review and meta-analysis found a strong relationship between $\dot{V}O_2$max and hemoglobin levels across a heterogenous pool of studies. Thus, these findings suggest that there is a robust relationship between $\dot{V}O_2$max and hemoglobin levels. These findings offer a comprehensive synthesis of the heterogeneous scientific corpus describing the relationship between hemoglobin and $\dot{V}O_2$max. These broad findings and simplified approach offer a foundation to chart future directions and testable hypotheses in this field.

## Supporting information

**S1 Checklist. PRISMA 2020 checklist.**
(PDF)

**S1 Fig. Association between hemoglobin concentration and maximal oxygen uptake ($\dot{V}O_2$max).** Bubble plot and meta-regression displaying the positive association between hemoglobin concentration and both absolute $\dot{V}O_2$max (A) and relative $\dot{V}O_2$max (B). Data for males are represented as purple bubbles, data for females are represented as green bubbles with plus sign symbols, and data for studies presenting males and females pooled (mixed) are represented as black bubbles with middle dot symbols. Each bubble represents a group from a single study and the size of bubbles represents the number of participants within the group. The solid line indicates the meta-regression line, and the dashed lines indicate the 95%

prediction interval associated with the meta-regression.
(DOCX)

**S2 Fig. Association between hematocrit and maximal oxygen uptake ($\dot{V}O_2$max).** Bubble plot and meta-regression displaying the positive association between hematocrit and both absolute $\dot{V}O_2$max (A) and relative $\dot{V}O_2$max (B). Data for males are represented as purple bubbles, data for females are represented as green bubbles with plus sign symbols, and data for studies presenting males and females pooled (mixed) are represented as black bubbles with middle dot symbols. Each bubble represents a group from a single study and the size of bubbles represents the number of participants within each group. The solid line indicates the meta-regression line, and the dashed lines indicate the 95% prediction interval associated with the meta-regression.
(DOCX)

**S1 Table. Risk of bias among observational articles.**
(DOCX)

**S2 Table. Risk of bias among interventional articles.**
(DOCX)

**S3 Table. Underlying data set.**
(XLSX)

## Acknowledgments

We thank the research teams and investigators for their rigorous efforts that contributed to each of the individual studies that made this review possible. Additionally, we thank members of the Human and Integrative Physiology and Clinical Pharmacology Laboratory at Mayo Clinic for intellectual discussions and feedback on earlier versions of the manuscript.

## Author Contributions

**Conceptualization:** Stephen A. Klassen, Chad C. Wiggins, Michael J. Joyner, Jonathon W. Senefeld.

**Data curation:** Kevin L. Webb, Ellen K. Gorman, Olaf H. Morkeberg, Stephen A. Klassen, Riley J. Regimbal, Chad C. Wiggins, Michael J. Joyner, Shane M. Hammer, Jonathon W. Senefeld.

**Formal analysis:** Kevin L. Webb, Jonathon W. Senefeld.

**Funding acquisition:** Michael J. Joyner, Shane M. Hammer, Jonathon W. Senefeld.

**Methodology:** Kevin L. Webb, Stephen A. Klassen, Shane M. Hammer, Jonathon W. Senefeld.

**Project administration:** Kevin L. Webb, Ellen K. Gorman, Shane M. Hammer.

**Supervision:** Stephen A. Klassen, Riley J. Regimbal, Chad C. Wiggins, Michael J. Joyner, Jonathon W. Senefeld.

**Visualization:** Kevin L. Webb, Ellen K. Gorman, Jonathon W. Senefeld.

**Writing – original draft:** Kevin L. Webb, Ellen K. Gorman, Shane M. Hammer, Jonathon W. Senefeld.

**Writing – review & editing:** Kevin L. Webb, Ellen K. Gorman, Olaf H. Morkeberg, Stephen A. Klassen, Riley J. Regimbal, Chad C. Wiggins, Michael J. Joyner, Shane M. Hammer, Jonathon W. Senefeld.

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
