## [Decision Letter · Decision Letter 0]

26 Dec 2022

PONE-D-22-29015The relationship between hemoglobin and V̇O2max: a systematic review and meta-analysisPLOS ONE

Dear Dr. Senefeld,

Thank you for submitting your manuscript to PLOS ONE. After careful consideration, we feel that it has merit but does not fully meet PLOS ONE’s publication criteria as it currently stands. Therefore, we invite you to submit a revised version of the manuscript that addresses the points raised during the review process.

The recommendations by the referees are opposite. Therefore, try your best to address all the referees' concerns. Please, do not assume that your manuscript will be accepted in the next round of revisions. Eventually, I may ask a third reviewer to make a decision. Good luck, Merry Christmas and Happy New Year!

We look forward to receiving your revised manuscript.

Kind regards,

Daniel Boullosa

Academic Editor

PLOS ONE

Journal Requirements:

Reviewers' comments:

Reviewer's Responses to Questions

**Comments to the Author**

1. Is the manuscript technically sound, and do the data support the conclusions?

Reviewer #1: No

Reviewer #2: Partly

2. Has the statistical analysis been performed appropriately and rigorously? 

Reviewer #1: No

Reviewer #2: Yes

3. Have the authors made all data underlying the findings in their manuscript fully available?

Reviewer #1: No

Reviewer #2: Yes

4. Is the manuscript presented in an intelligible fashion and written in standard English?

Reviewer #1: Yes

Reviewer #2: Yes

5. Review Comments to the Author

Reviewer #1: The authors present a systematic review with meta-analysis on the relationship between changes in haemoglobin with changes in maximal oxygen consumption. This is an important topic as the ability to carry oxygen in the blood has been shown to influence maximal oxygen consumption. I have a few major concerns that I listed below.

1. I do not believe that it is appropriate to combine interventions simply because they fall under the same category (i.e., all dietary interventions). This would be like combining all pharmacological interventions simply because they are pharmaceuticals. The external validity of these results would be quite limited. Separate analysis for each sub-category would be required to limit clinical heterogeneity.

2. The authors did not describe the differences/limitations in the methods used to measure haemoglobin. The reliability and validity for measures of haemoglobin concentration are very low considering it is strongly affected by changes in plasma volume. Haemoglobin concentration is the gold standard as it is both stable and valid.

3. The authors combined studies that included healthy populations with clinical populations. This is not recommended as it may introduce significant clinical heterogeneity. Why did they not conduct a stratified meta-analysis of these groups?

4. I have concerns regarding the analysis of risk of bias of individual studies. I do not believe I have ever seen a systematic review that found that the risk of bias for every category for all intervention studies was ‘low’. Given the large number of studies included in the analysis (158 intervention studies), this would be virtually impossible.

5. The statistical methods used to conduct the meta-regressions of the percentage improvement is unclear. Based on the available information, it appears that they used the pooled standard deviation (i.e., combing the baseline standard deviation (SD) with the follow up SD). This would be inappropriate since the SD of the change score is not the same as the pooled SD. The SD of the change score is rarely provided in studies. There are ways to estimate to this value; however, none are described in the methodology.

6. I am unsure how the authors determined the causes of heterogeneity since there are no results that describe that statistical heterogeneity in the manuscript. This not only introduces significant bias in the results but limits the external validity of the findings altogether.

Reviewer #2: I read with great interest and pleasure the article "The relationship between haemoglobin and V̇O2max: a systematic review and meta-analysis". The manuscript reflects the magnitude of the task the authors have carried out. From the methodological point of view of a meta-analysis and a systematic review, the manuscript meets the formal and quality conditions usually required for publication in a prestigious journal.

Conducting a systematic review or meta-analysis on the relationship between haemoglobin and VO2max is very difficult if practical conclusions are to be derived, because haemoglobin is only one of the possible factors that can limit/determine VO2max. Studying the relationship between only one of the determinants and the VO2max result has the risk that some readers may not fully understand the real meaning of the relationship or non-relationship of these two variables.

I think it would be advisable to make a minor change, consisting of a more concrete expression of the research question(s), in order to facilitate the understanding of the development of the manuscript and to make the answer(s) to the question(s) more explicitly concrete.

I suggest that the authors, beyond the meta-analysis or systematic review, explicitly state the limitations of the interpretation of the studied relationship:

a) The change in haemoglobin in intervention studies, not only involve changes in haemoglobin concentration (mass), but often involve other added changes that can also influence VO2max such as a change in blood volume or cardiac output, for example.

b) The differences between recipients and donors of blood are not only due to the change in the amount of haemoglobin in the donor or recipient.

c) The summary of the articles used in the systematic review and meta-analysis are based on the interpretation of statistical significance (p-value), but I find no interpretation of the small magnitude of change found in many of the articles reviewed, which are below the minimum detectable change in VO2max determination, and well below the minimum clinically significant change. So the changes are less important than apparently suggested or concluded. This point seems to me to be particularly important.

d) In the different intervention studies, the changes found are not always comparable according to the duration of the intervention (acute/chronic) or time of measurement after the end of the intervention (hours, days, weeks...).

e) The methodological limitations of the observational studies used (population variety, methodological variety, ....) are not clearly highlighted.

f) I believe that the conclusions should be improved. If in the introduction they state that there is a consensus on the relationship between haemoglobin and VO2 (transport), to what uncertainties in knowledge on the subject did the authors intend to respond with their review and meta-analysis?

g) I believe that it does not follow from their study that the improvement in performance (they have not studied it) caused by the increase in haemoglobin is an argument for the prohibition of blood doping. It should be banned because it is an artificial method, because it is a potentially very dangerous method, and because it increases performance, not necessarily because of the increase in haemoglobin.

h) Finally, I believe that the bibliographical references are abused, and there are a good number of them that could be removed without altering the quality of the study. For example, I do not think it is necessary to use 6 articles to support a clear statement (lin 42), one would be sufficient. Furthermore, some references are not very relevant in the context in which they have been used, such as ref. 23, which does not study the VO2-Hb ratio, but is a review of anaemia in athletes (lin 47). I also fail to understand the use of reference 35 to justify that above 500m altitude can be considered a hypoxic-hypobaric environment, nor that FIO2 is less than 20%, nor do I agree with that statement, nor do I believe that such a statement follows from reference 35.

I submit these observations for your consideration, which I believe will make your manuscript more useful, and I hope they will be addressed (if you deem it appropriate) or responded to in a reasoned manner.

In any case, I would like to congratulate you on your work.

6. PLOS authors have the option to publish the peer review history of their article (what does this mean?). If published, this will include your full peer review and any attached files.

Reviewer #1: No

Reviewer #2: **Yes: **José Antonio de Paz

---

## [Author Response · Author response to Decision Letter 0]

28 Feb 2023

Academic Editor

Thank you for submitting your manuscript to PLOS ONE. After careful consideration, we feel that it has merit but does not fully meet PLOS ONE’s publication criteria as it currently stands. Therefore, we invite you to submit a revised version of the manuscript that addresses the points raised during the review process.

Response: Thank you for the opportunity to revise this manuscript and respond to comments from expert reviewers. The comments from the reviewers enhanced the quality and potential impact of this manuscript, and the comments have been responded to accordingly.

Journal Requirements:

Response: Done. We have ensured that the revised manuscript meets applicable journal requirements.

Response: Thank you for highlighting this omission. This work was supported by the National Heart, Lung, and Blood Institute T-32-HL105355 (K.L.W.), R-35-HL139854 (C.C.W. and M.J.J.), and F-32-HL131151 (J.W.S.). 

Response: Done. We have now included the study’s minimal data set as a supplementary file (Table S3) and specified where this minimal data set may be found.

 

Reviewer 1

The authors present a systematic review with meta-analysis on the relationship between changes in haemoglobin with changes in maximal oxygen consumption. This is an important topic as the ability to carry oxygen in the blood has been shown to influence maximal oxygen consumption. I have a few major concerns that I listed below.

Response: Thank you for the detailed review of the manuscript and productive review. The constructive and insightful comments of the reviewer were prudently incorporated into the manuscript and have improved the quality and potential impact of this work.

1. I do not believe that it is appropriate to combine interventions simply because they fall under the same category (i.e., all dietary interventions). This would be like combining all pharmacological interventions simply because they are pharmaceuticals. The external validity of these results would be quite limited. Separate analysis for each sub-category would be required to limit clinical heterogeneity.

Response: We agree and as suggested by the reviewer, have now added separate analyses for each sub-category. In accordance with Cochrane suggested methodology, a minimum of two studies are needed to perform meta-analyses (please see line 174), limiting several sub-category analyses described below. This approach suggested by the reviewer likely enhances external validity and limits clinical heterogeneity. In an effort of transparent scientific reporting, we also present our original, pooled analyses. For clarity, we have described the categorical separation of categories and sub-categories below.

1. Observational compared to interventional designs. Maintaining this important stratification from the original submission, we continue to separate analyses between observational data and interventional data. In this context, the remaining categories and subcategories are described below.

2. Blood transfusion and blood donation. We have maintained pooled and stratified analyses from the original submission— stratified analyses included 11 studies associated with blood transfusion and 17 studies associated with blood donation. In this context, this section of the text remains unchanged.

3. Erythropoiesis stimulating agents. Our original pooled analysis represented 11 studies examining erythropoietin supplementation and one study examining xenon inhalation. In the revised manuscript, we have stratified our analyses by two different erythropoiesis stimulating agents— erythropoietin and xenon. Please see lines 357-359. Due to an insufficient number of studies (only 1 study), stratified analyses were not performed for xenon inhalation alone.

4. Dietary supplementation. There was a broad spectrum of heterogenous substances used for dietary supplementation. In this context and as now noted in the text (please see lines 378-379), iron and echinacea supplementation were the only sub-categories with an adequate number of studies needed to perform meta-analyses. This important analysis has been added to the text, please see lines 386–392.

5. Environmental hypoxia. Environmental hypoxia analyses pooled two sub-categories— high-altitude environments and normobaric hypoxia. In this framework, the revised manuscript includes analyses stratified into these two sub-categories (high altitude acclimatization and sojourn vs. chronic normobaric hypoxia). Please see lines 409–414.

6. Aerobic (de)training. We have maintained pooled and stratified analyses from the original submission — stratified analyses included 23 studies associated with aerobic training and 7 studies associated with aerobic detraining. In this context, this section of the text remains unchanged.

2. The authors did not describe the differences/limitations in the methods used to measure haemoglobin. The reliability and validity for measures of haemoglobin concentration are very low considering it is strongly affected by changes in plasma volume. Haemoglobin concentration is the gold standard as it is both stable and valid.

Response: Thank you for the keen suggestion, we now describe the differences/limitations in the methods used to measure hemoglobin concentration and hemoglobin mass in the main text of the revised manuscript, in both the methods (please see lines 113-125) and the limitations section (please see lines 513-520). Notably, our analyses of the relationship between hemoglobin levels and VO2max were performed in duplicate, using both hemoglobin concentration and hemoglobin mass independently. The results of these analyses are highly convergent which provide ecological validity and clinical generalizability to our findings.

3. The authors combined studies that included healthy populations with clinical populations. This is not recommended as it may introduce significant clinical heterogeneity. Why did they not conduct a stratified meta-analysis of these groups?

Response: The reviewer’s point is well taken and has been incorporated into the revised manuscript, as applicable— for example, please see lines 233-236. Notably, however, articles that reported data for participants with disordered erythropoiesis represented a small portion of the articles included in our analyses— only about 2% (10 of 384 articles). As only a small number of studies enrolled participants with disordered erythropoiesis, it was not plausible to stratify many analyses, specifically among interventional studies. Instead, observational analyses were further stratified for healthy and clinical populations separately (please see lines 257-263). 

More broadly, the goal of our work was to examine the relationship between hemoglobin levels and V̇O2max, and we intentionally attempted to restrict our analyses to people that may be generally considered as healthy. In this context, our a priori methodology specified the inclusion of clinical populations that exhibit alterations in hemoglobin levels without secondary pathologies and which may otherwise be considered as ‘healthy’. As such, the authors carefully screened participants of individual studies and excluded groups of ‘clinical populations’ who had secondary pathologies. This screening process is represented in the PRISMA diagram (Figure 1), which shows that 69 studies were excluded due to ineligible participant population. 

4. I have concerns regarding the analysis of risk of bias of individual studies. I do not believe I have ever seen a systematic review that found that the risk of bias for every category for all intervention studies was ‘low’. Given the large number of studies included in the analysis (158 intervention studies), this would be virtually impossible.

Response: The reviewer’s point is well taken and was actively discussed. We acknowledge that the original standards used to assess the risk of bias may have been too lenient. Thus, the risk of bias was reassessed with a more stringent interpretation, and the tables have been updated accordingly (see tables S1 & S2). Risk assessment is now also explicitly described in the text — please see lines 237-247.

However, it is also important to note that our a priori approach included rigorous screening and exclusion criteria which resulted in the exclusion of many studies with a probable risk of bias. As evidenced in Figure 1, 589 articles were excluded due to ineligible assessments for hemoglobin (386 articles) or VO2max values (134 articles) and ineligible participant populations (69 articles). Thus, our a priori approach likely contributed to reduced risk of bias among studies included in analyses.

5. The statistical methods used to conduct the meta-regressions of the percentage improvement is unclear. Based on the available information, it appears that they used the pooled standard deviation (i.e., combing the baseline standard deviation (SD) with the follow up SD). This would be inappropriate since the SD of the change score is not the same as the pooled SD. The SD of the change score is rarely provided in studies. There are ways to estimate to this value; however, none are described in the methodology.

Response: Thank you. We have amended the methods in an effort to improve clarity — please see lines 208-213. We agree with the strong rationale presented by the reviewer: 

1) standard deviation would be needed to perform meta-analyses of dependent data regarding the percentage of improvement in VO2max.

2) many studies did not report the standard deviation of the percent change in VO2max with the applied intervention. 

For these reasons, we performed linear regressions (and not meta-regressions) to examine data regarding the percentage of improvement in VO2max. Our initial approach was in line with the reviewer’s suggestion, and the methods have been amended to more clearly define this methodology.

6. I am unsure how the authors determined the causes of heterogeneity since there are no results that describe that statistical heterogeneity in the manuscript. This not only introduces significant bias in the results but limits the external validity of the findings altogether.

Response: Thank you for pointing out this important limitation, and we apologize for the omission in the previous manuscript. We have now added calculations of tau-squared (τ2) as an estimate of between-study heterogeneity within appropriate findings. Please find amendments to the presentation of results throughout the manuscript.

 

Reviewer 2

I read with great interest and pleasure the article "The relationship between haemoglobin and V̇O2max: a systematic review and meta-analysis". The manuscript reflects the magnitude of the task the authors have carried out. From the methodological point of view of a meta-analysis and a systematic review, the manuscript meets the formal and quality conditions usually required for publication in a prestigious journal.

Response: Thank you for the many constructive suggestions and supportive tone of the review. These excellent comments have served to improve the quality of this manuscript.

Conducting a systematic review or meta-analysis on the relationship between haemoglobin and VO2max is very difficult if practical conclusions are to be derived, because haemoglobin is only one of the possible factors that can limit/determine VO2max. Studying the relationship between only one of the determinants and the VO2max result has the risk that some readers may not fully understand the real meaning of the relationship or non-relationship of these two variables.

I think it would be advisable to make a minor change, consisting of a more concrete expression of the research question(s), in order to facilitate the understanding of the development of the manuscript and to make the answer(s) to the question(s) more explicitly concrete.

Response: Thank you for the keen suggestion. We have amended the abstract and conclusion to enhance the expression of the research question—please see lines 5-10 and 525-528. The overall goal of this work was to provide a comprehensive systematic review and meta-analysis of the relationship between hemoglobin levels and V̇O2max in humans from the diverse scientific literature. As the reviewer suggested, there are many putative confounding variables that may limit/determine V̇O2max in humans, thus, and we believe this work may offer a foundation to chart future directions and testable hypotheses in this field.

I suggest that the authors, beyond the meta-analysis or systematic review, explicitly state the limitations of the interpretation of the studied relationship:

Response: Thank you. We agree and have added a limitations section to the manuscript to highlight the limitations of our simplistic analysis clearly and explicitly— please see lines 496-520.

a) The change in haemoglobin in intervention studies, not only involve changes in haemoglobin concentration (mass), but often involve other added changes that can also influence VO2max such as a change in blood volume or cardiac output, for example.

Response: The reviewer’s point is well taken and was actively discussed. We added a description of some of the potential influence of changes in blood volume and cardiac output in the discussion— please see lines 477-479. Considering the reviewer’s comment, we also further emphasized this topic within newly added limitations section, please see lines 507-512.

b) The differences between recipients and donors of blood are not only due to the change in the amount of haemoglobin in the donor or recipient.

Response: We agree with the reviewer. Due to the physiological differences in blood donation versus transfusion, stratified analyses were performed. Furthermore, we have amended the discussion to clarify this important point. Please see lines 477-479.

c) The summary of the articles used in the systematic review and meta-analysis are based on the interpretation of statistical significance (p-value), but I find no interpretation of the small magnitude of change found in many of the articles reviewed, which are below the minimum detectable change in VO2max determination, and well below the minimum clinically significant change. So the changes are less important than apparently suggested or concluded. This point seems to me to be particularly important.

Response: Within the interventional analyses, some studies report a large change in VO2max, yet others report a small, or ‘negligible’ change in VO2max. As the reviewer has mentioned below in comment d), this heterogeneity of findings may be due to differences in duration of the intervention, magnitude of intervention (i.e., dose or amount of blood transfused), or the time of measurement after the end of the intervention. We have added this key concept in the limitations section of the manuscript— please see lines 500-507. Within the meta-analyses stratified by the performed intervention, the pooling of study-level data provides a cumulative finding within the area of interest.

Small, or ‘negligible’ changes in VO2max or hemoglobin concentration may be a benefit in the presented regressions by providing a ‘physiological anchor’ to an intervention without physiological alterations. For instance, Figure 4 denotes the findings from heterogenous studies, some with small changes in VO2max and others with large changes in VO2max. This heterogeneity may be critical to the performed meta-regressions and linear regressions. Similarly, if all studies showed a large, unidirectional change in hemoglobin levels and VO2max, we may not be able to assess a ‘slope’ given that no values near zero would provide directionality.

d & e) In the different intervention studies, the changes found are not always comparable according to the duration of the intervention (acute/chronic) or time of measurement after the end of the intervention (hours, days, weeks...).

The methodological limitations of the observational studies used (population variety, methodological variety, ....) are not clearly highlighted.

Response to d) and e): These points are well-received. In response to these important considerations, we have added these concepts to a new limitations section. Please see lines 497-512.

f) I believe that the conclusions should be improved. If in the introduction they state that there is a consensus on the relationship between haemoglobin and VO2 (transport), to what uncertainties in knowledge on the subject did the authors intend to respond with their review and meta-analysis?

Response: This work aimed to provide a comprehensive review of the diverse scientific corpus. One example of uncertainty is presented in lines 62-68. Moreover, this work may help identify further areas of uncertainty and stimulate research to better our understanding of the physiological consequences of interventional manipulation of hemoglobin levels. By creating an unbiased synthesis of this relationship (between hemoglobin levels and V̇O2max in humans), we believe our work may serve as a foundation to chart future directions and testable hypotheses in this field. The conclusion has been amended to better describe the aim of this work. Please see lines 525-528.

g) I believe that it does not follow from their study that the improvement in performance (they have not studied it) caused by the increase in haemoglobin is an argument for the prohibition of blood doping. It should be banned because it is an artificial method, because it is a potentially very dangerous method, and because it increases performance, not necessarily because of the increase in haemoglobin.

Response: Thank you for your comment. We have removed this sentence.

h) Finally, I believe that the bibliographical references are abused, and there are a good number of them that could be removed without altering the quality of the study. For example, I do not think it is necessary to use 6 articles to support a clear statement (lin 42), one would be sufficient. Furthermore, some references are not very relevant in the context in which they have been used, such as ref. 23, which does not study the VO2-Hb ratio, but is a review of anaemia in athletes (lin 47). I also fail to understand the use of reference 35 to justify that above 500m altitude can be considered a hypoxic-hypobaric environment, nor that FIO2 is less than 20%, nor do I agree with that statement, nor do I believe that such a statement follows from reference 35.

Response: Our apologies, we certainly understand the reviewer’s point. Although our intention was to pay homage to the many luminaries in this field, we understand how this was construed as abusing bibliographical references. In this context, we have removed several citations in the introduction (n=8) and discussion (n=6). Additionally, we have revised the methodology to more accurately denote the environmental conditions eligible for the determination of VO2max. Please see lines 110-112.

I submit these observations for your consideration, which I believe will make your manuscript more useful, and I hope they will be addressed (if you deem it appropriate) or responded to in a reasoned manner.

In any case, I would like to congratulate you on your work.

Response: Many thanks. The comments from the reviewers represent a significant investment of time and effort, and we believe the comments enhanced the quality and potential impact of this manuscript.

---

## [Decision Letter · Decision Letter 1]

13 Apr 2023

PONE-D-22-29015R1The relationship between hemoglobin and V̇O2max: a systematic review and meta-analysisPLOS ONE

Dear Dr. Senefeld,

Thank you for submitting your manuscript to PLOS ONE. After careful consideration, we feel that it has merit but does not fully meet PLOS ONE’s publication criteria as it currently stands. Therefore, we invite you to submit a revised version of the manuscript that addresses the points raised during the review process.

 Please, address, the best you can, the reviewers' suggestions. He is a true expert on this topic and on the methodology of a systematic review.

We look forward to receiving your revised manuscript.

Kind regards,

Daniel Boullosa

Academic Editor

PLOS ONE

Reviewers' comments:

Reviewer's Responses to Questions

**Comments to the Author**

1. If the authors have adequately addressed your comments raised in a previous round of review and you feel that this manuscript is now acceptable for publication, you may indicate that here to bypass the “Comments to the Author” section, enter your conflict of interest statement in the “Confidential to Editor” section, and submit your "Accept" recommendation.

Reviewer #1: (No Response)

Reviewer #2: All comments have been addressed

2. Is the manuscript technically sound, and do the data support the conclusions?

Reviewer #1: Partly

Reviewer #2: Yes

3. Has the statistical analysis been performed appropriately and rigorously? 

Reviewer #1: No

Reviewer #2: Yes

4. Have the authors made all data underlying the findings in their manuscript fully available?

Reviewer #1: No

Reviewer #2: Yes

5. Is the manuscript presented in an intelligible fashion and written in standard English?

Reviewer #1: Yes

Reviewer #2: Yes

6. Review Comments to the Author

Reviewer #1: GENERAL

I would like to start by commending the authors for addressing my initial feedback. They have addressed significant methodological issues. Below I comment on several specific issues that should help to improve the overall quality of their review.

Comment 1

Please use the headings listed in the PRISMA checklist. Further, there are sub-headings that should be used to organize each of the sections of the review. This will improve both the transparency and flow.

Details with a full guidance is available in the second reference below.

References

Page MJ, McKenzie JE, Bossuyt PM, Boutron I, Hoffmann TC, Mulrow CD, et al. The PRISMA 2020 statement: an updated guideline for reporting systematic reviews. BMJ. 2021;372:n71.

Page MJ, Moher D, Bossuyt PM, Boutron I, Hoffmann TC, Mulrow CD, et al. PRISMA 2020 explanation and elaboration: updated guidance and exemplars for reporting systematic reviews. BMJ. 2021;372:n160.

INTRODUCTION

Comment 2

Page 4, line 39: It is important to describe how a VO2max is assessed and what physiological measurements constitutes a VO2max. It would also be beneficial to explain the difference between VO2max and VO2peak. I am assuming that the authors combined studies that included both measurements.

Comment 3

Page 4, line 50: This section should be removed from the introduction. It may be more appropriate in the methods section to explain the eligibility criteria, or more importantly in the limitations section of the review.

Comment 4

Page 4, line 60: “Although a strong …”. If we already know that there is a strong relationship between haemoglobin levels and VO2max, why conduct this review? Maybe acknowledge all the work that has been done in the past then go right into the lack or potential lack of systematic reviews with meta-analysis. The authors are on the right track by need to strengthen their rationale. It may also be beneficial to acknowledge previous systematic reviews on the topic, some of the limitations in those reviews, and how this review differs (e.g., addresses previous limitations, looks at other outcomes or interventions).

Comment 5

Page 5, line 74: “Moreover…” This is a secondary objective. The authors use both specific (sex) and general (interventions) examples. First, it is important to state that there is a secondary objective. Second, it may be better to explain that you will conduct secondary analyses to determine the influence of ‘population characteristics’ (sex, age, etc.) and ‘intervention characteristics’. Specific examples of stratified and meta-regression should be provided in the Methods (i.e., how you will address statistical heterogeneity).

METHODS

Comment 6

Please list and describe effect measures that will be used to describe the results. I believe that some of this data is in the “Statistical analysis” section. This should be in a section called “Effect Measures”.

Comment 7

Have you considered using p-values to calculate the missing standard deviation by using a dependent t-test? This could allow you to conduct meta-regressions (albeit a more conservative analysis when p-values or only listed as p < 0.05). I would consider a p-value of “<0.05” as p = 0.0499 if using a t-test to calculate a SD, though any assumptions should be listed.

Comment 8

All regressions that do not consider the within and between group variance must be removed from the results. It is not appropriate to pool studies without considering the weight of each study as the results will mislead the reader. Practitioners use meta-analyses to make decisions about patient care so they must be conducted accurately.

Comment 9

Page 6, line 88: Why was only a single database used to conduct the literature search? Are you concerned that you have important missed studies?

Comment 10

Page 8, line 149: Why was data that included confidence intervals as a measure of uncertainty excluded when transformations can easily be performed, especially since this was does for SEM to SD.

Comment 11

Page 8, line 149: If data points that were only expressed as means without associated error, why were studies included in a non-weighted regression?

Comment 12

Page 8, line 151: This section should be in the section ‘Data Items’ as described in PRISMA. The items should follow PICOS.

Comment 13

Page 9, line 169: Statistical Analysis should be divided into Effect Measures, Synthesis Methods, and Reporting Bias Assessment, as indicated by PRISMA.

Comment 14

Page 9, line 173: “Analyses pooled outcome data…”. This sentence does not make sense. The analyses didn’t pool the data, the authors did. Second, it is stated that data was pooled from each study but the sentence before states that they were pooled by study design.

Comment 15

I think sensitivity analyses should be performed for: 1. Relationship for haemoglobin with VO2peak vs VO2max, 2. Study design.

Comment 16

Page 10, line 176: Can you please use Higgin’s I^2 statistic to report the statistical heterogeneity. It does have limitations (as does Q and tau^2), but it is a measure that explains the relative statistical heterogeneity and is also easier for the reader to understand (especially those that do not understand meta-analysis)

Comment 17

How did you account for small sample bias or publication bias? I would suggest Egger’s for small sample size and Vevea and Hedges Weight-Function Model for Publication Bias.

RESULTS

Comment 18

All results should be described in past tense.

Comment 19

Please use the PRISMA headings as mentioned above.

Comment 20

If stratified analyses were performed (e.g., by sex: combined, male, female), please provide results that describe if there were significance (p-value) differences for each analysis (either from the initial analysis or between (among) stratified groups. This should be completed for all results in text and in tables.

Comment 21

Page 16, line 314: Why was the study removed? Was it the ‘only’ one study? This could provide a rationale for the removal, but needs to be explained to the reader.

DISCUSSION

Comment 24

General: Please use or at least try to address PRISMA sub-headings. 1. General interpretation of results, 2. Limitations of evidence included in the review, 3. Limitations of review process, 4. Implications for practice, policy and future research.

Comment 25

Page 24, line 497: What do you mean by simplistic design?

Comment 26

Page 24, line 499: What are you referring to by “more complex statistical methods”? You can use aggregate data to perform multivariate analyses. Limitations in the complexity of statistical analyses are probably due to the software that used to conduct the analyses. Software packages available in ‘R’ make it very easy to conduct multivariate analyses. The primary (and albeit significant) limitation using aggregate data is the fallacy of regression to the mean, which occurs when using continuous data of population characteristics to perform meta-regressions. However, this limitation occurs regardless of the type of analyses (i.e., univariate or multivariate analyses).

Comment 27

Page 24, line 504: “For example…” This point is only a limitation if you choose not to run stratified or meta-regressions on intervention characteristics, which were completed for some variables. You can say that the limitations, specifically with your review was that you chose not to run these analyses for which there may be valid reasons which should be stated (e.g., the data was not available, etc.). But I would explain why you chose not examine the variables listed here. However, it is not a limitation to the approach as it is best to include all studies and then find out the source of statistical heterogeneity by means of the clinical heterogeneity.

Comment 28

Page 25, line 513: Good point. Though the primary differences in haemoglobin would be concentration versus mass. Please cite the literature for your examples. If you believe that your results show that there was no difference between the two measurement types, please explain this to the reader in further detail as it is an important finding that conflicts with current literature.

CONCLUSION

Comment 29

Page 25, line 522: The results suggest that the pooled effect measure changed after addressing measures of clinical heterogeneity (though, it was not stated in the results section of the differences in modifiers were considered to be significant (no p-value to accompany the change in variance). Therefore, given the available results, I am not sure if this statement is true.

Comment 30

Page 25, line 525: I would argue that there is still bias in the methods used to conduct the synthesis of the results. However, there are efforts within the manuscript to limit the degree of bias.

Reviewer #2: I have reviewed the document submitted by the authors again and have observed that the changes made in this second manuscript are minimal compared to the first, except for some statistical treatments (heterogeneity, Tau). The observations made in the first round are mainly included in a broad section of limitations that have been added to this second document.

The fundamental objections remain present in this second manuscript. Nevertheless, considering the magnitude of the effort made, the number of articles reviewed, and the fact that the limitations of the study are now explicitly stated based on the observations made, I believe that the article can be published. I think it is sufficiently clear what the real scope of the work's conclusions may be for non-expert readers in the field.

7. PLOS authors have the option to publish the peer review history of their article (what does this mean?). If published, this will include your full peer review and any attached files.

Reviewer #1: No

Reviewer #2: **Yes: **José Antonio de Paz

---

## [Author Response · Author response to Decision Letter 1]

1 Aug 2023

Academic Editor

Thank you for submitting your manuscript to PLOS ONE. After careful consideration, we feel that it has merit but does not fully meet PLOS ONE’s publication criteria as it currently stands. Therefore, we invite you to submit a revised version of the manuscript that addresses the points raised during the review process.

Response: Thank you for spearheading this high-quality, collegial, and productive review process. As we described previously, the comments from the reviewers enhanced the quality and potential impact of this manuscript.

Please, address, the best you can, the reviewers' suggestions. He is a true expert on this topic and on the methodology of a systematic review.

Response: We read with great interest the comments of the reviewer and thank them for the efforts to improve our manuscript. We appreciate the many constructive suggestions.

Reviewer 1

I would like to start by commending the authors for addressing my initial feedback. They have addressed significant methodological issues. Below I comment on several specific issues that should help to improve the overall quality of their review.

Response: Although not the intent, we recognize the following comments may be interpreted as gratuitously flattering. We would like to thank the reviewer for the thorough and constructive review of this manuscript. The comments from the reviewer represent a significant investment of time and effort, and we believe the comments enhanced the quality and potential impact of this manuscript. Thank you.

Please also note that throughout the response to reviewers, specific text excerpts are referenced using line numbers— these line numbers are associated with the redlined version of the manuscript. 

Comment 1 | Please use the headings listed in the PRISMA checklist. Further, there are sub-headings that should be used to organize each of the sections of the review. This will improve both the transparency and flow.

Details with a full guidance is available in the second reference below.

References

Page MJ, McKenzie JE, Bossuyt PM, Boutron I, Hoffmann TC, Mulrow CD, et al. The PRISMA 2020 statement: an updated guideline for reporting systematic reviews. BMJ. 2021;372:n71.

Page MJ, Moher D, Bossuyt PM, Boutron I, Hoffmann TC, Mulrow CD, et al. PRISMA 2020 explanation and elaboration: updated guidance and exemplars for reporting systematic reviews. BMJ. 2021;372:n160.

Response: Done. We have changed the headings (Level 1 Headings) and sub-headings (Level 2 Headings) to align with the headings and sub-headings associated with the PRISMA checklist. Please find these new headings informed by the PRISMA checklist throughout the manuscript. For additional organization for readers, we have also provided Level 3 Headings as appropriate. Notably, however, the PRISMA checklist does not (explicitly) offer suggestions for Level 3 Headings. In this context, we discussed Level 3 Headings intensively within our author team and included headings as warranted. 

INTRODUCTION

Comment 2 | Page 4, line 39: It is important to describe how a VO2max is assessed and what physiological measurements constitutes a VO2max. It would also be beneficial to explain the difference between VO2max and VO2peak. I am assuming that the authors combined studies that included both measurements.

Response: Thank you for pointing out these keen points. We have included additional contextual background on V̇O2max testing within the manuscript (please see lines 110 to 115) and responded to your comments below. 

We have considered your comment as two separate points. First, the standardized assessment of V̇O2max is described in the methods under the “Eligibility criteria” sub-heading— please see lines 102 to 122. We have added additional text to clarify the criteria used to qualify measurements as V̇O2max (as opposed to V̇O2peak). 

Second, as also described in the methods under the “Eligibility criteria sub-heading”, the authors — in accordance with our a priori protocol — discussed the inclusion of V̇O2peak measurements intensively before data collection commenced and decided to only include measurements that represent V̇O2max. In this framework, studies that assessed V̇O2peak or estimated V̇O2max were not eligible for inclusion.

Comment 3 | Page 4, line 50: This section should be removed from the introduction. It may be more appropriate in the methods section to explain the eligibility criteria, or more importantly in the limitations section of the review.

Comment 4 | Page 4, line 60: “Although a strong …”. If we already know that there is a strong relationship between haemoglobin levels and VO2max, why conduct this review? Maybe acknowledge all the work that has been done in the past then go right into the lack or potential lack of systematic reviews with meta-analysis. The authors are on the right track by need to strengthen their rationale. It may also be beneficial to acknowledge previous systematic reviews on the topic, some of the limitations in those reviews, and how this review differs (e.g., addresses previous limitations, looks at other outcomes or interventions).

Response: We have elected to respond to comments 3 and 4 from the reviewer – which both correspond to the same section of the manuscript—using one response. We appreciate the reviewer’s perspectives and have more aptly entitled this section of the manuscript as “Rationale”. In accordance with the description of the rationale section provided in the PRISMA checklist – describe the rationale for the review in the context of existing knowledge – we have described our primary rationale within the context of existing knowledge. While we agree with the reviewer that this information may also be appropriate in the methods or limitations section, this information describes the rationale for the review and continues to be within the introduction section.

As described in the text, although many studies have been performed on the relationship between hemoglobin levels and V̇O2max, there are ambiguous areas associated with biological diversity and heterogenous study designs, and a comprehensive synthesis of information may provide key insights from the scientific corpus.

Comment 5 | Page 5, line 74: “Moreover…” This is a secondary objective. The authors use both specific (sex) and general (interventions) examples. First, it is important to state that there is a secondary objective. Second, it may be better to explain that you will conduct secondary analyses to determine the influence of ‘population characteristics’ (sex, age, etc.) and ‘intervention characteristics’. Specific examples of stratified and meta-regression should be provided in the Methods (i.e., how you will address statistical heterogeneity).

Response: Done. We agree and have added explicit acknowledgement that there is a secondary objective. Additionally, we included that the secondary analyses were conducted to determine the influence of population characteristics and intervention characteristics—both with inclusion of specific examples. Please find this revised section on lines 76 to 79.

METHODS

Comment 6 | Please list and describe effect measures that will be used to describe the results. I believe that some of this data is in the “Statistical analysis” section. This should be in a section called “Effect Measures”.

Response: Done. As requested, we have listed and described effect measures that are used to describe the results. This important information may now be found under the “Effect measures” subheading—please see lines 181 to 187.

Comment 7 | Have you considered using p-values to calculate the missing standard deviation by using a dependent t-test? This could allow you to conduct meta-regressions (albeit a more conservative analysis when p-values or only listed as p < 0.05). I would consider a p-value of “<0.05” as p = 0.0499 if using a t-test to calculate a SD, though any assumptions should be listed.

Response: We did consider this important point from the reviewer. However, no paper provided means and p-values without estimate of error. In this context, this suggested data transformation was not warranted. Please find that we have removed this section from the text—see lines 159 to 161.

Comment 8 | All regressions that do not consider the within and between group variance must be removed from the results. It is not appropriate to pool studies without considering the weight of each study as the results will mislead the reader. Practitioners use meta-analyses to make decisions about patient care so they must be conducted accurately.

Response: Thank you for the comment. We agree with the reviewer that there are limitations in using regressions that do not consider within group variance and between group variance. Within this manuscript, all regressions performed considered between group variance. However, applied linear regressions do not account for within group variance due to each data point considered as a mean without measurement of standard error. The linear regressions performed are limited by the available data and this limitation has been appropriately caveated within the methods (please see lines 226 to 233). When transforming the data to examine the effects of manipulating hemoglobin levels on changes in V̇O2max, no manuscripts report the percent change with associated standard error. We have added this topic as a limitation in the discussion (please see lines 524 to 542). Although these linear regressions have limitations, we have elected to keep these analyses as they provide an important comparison with other seminal work in the field [1,2].

[1] Calbet JA, Lundby C, Koskolou M, Boushel R. Importance of hemoglobin concentration to exercise: acute manipulations. Respir Physiol Neurobiol. 2006 Apr 28;151(2-3):132-40. doi: 10.1016/j.resp.2006.01.014. Epub 2006 Mar 3. PMID: 16516566.

[2] Ekblom BT. Blood boosting and sport. Baillieres Best Pract Res Clin Endocrinol Metab. 2000 Mar;14(1):89-98. doi: 10.1053/beem.2000.0056. PMID: 10932813.

Comment 9 | Page 6, line 88: Why was only a single database used to conduct the literature search? Are you concerned that you have important missed studies?

Response: During our preliminary literature searches, we were deluged with high quality scientific articles. During preliminary searches, we also found significant duplication between databases. Thus, after preliminary searches, we elected to use a single database to conduct this literature search. This important limitation has been added to the limitations section – please see lines 524 to 525. After carefully searching references of all included articles for additional inclusions, we only found three additional studies to include. In this context, we are not concerned that we have important missed studies, and we were very pleased to have meta-analyzed data from nearly 400 studies.

Comment 10 | Page 8, line 149: Why was data that included confidence intervals as a measure of uncertainty excluded when transformations can easily be performed, especially since this was does for SEM to SD.

Response: Thank you for this comment. We looked back through our exclusion criteria for each study and found that no study was excluded on the basis of confidence intervals as a measure of uncertainty. Thus, we have removed this excerpt from the manuscript— please find strike-through text on lines 159 to 161.

Comment 11 | Page 8, line 149: If data points that were only expressed as means without associated error, why were studies included in a non-weighted regression?

Response: Good point. As discussed in response to Comment 8, we did not include data points that were expressed as means without associated error in meta-regression analyses. However, applied linear regressions were performed and presented using means without associated error, and these analyses have been appropriately caveated throughout the text. 

Comment 12 | Page 8, line 151: This section should be in the section ‘Data Items’ as described in PRISMA. The items should follow PICOS.

Response: These changes have been made as requested by the reviewer, and the section entitled ‘Data Items’ follows PICOS. Please see lines 143 to 161.

Comment 13 | Page 9, line 169: Statistical Analysis should be divided into Effect Measures, Synthesis Methods, and Reporting Bias Assessment, as indicated by PRISMA.

Response: Done. As suggested by the reviewer and indicated by PRISMA, the statistical analysis section has been divided into Effect Measures, Synthesis Methods, and Reporting Bias Assessment. Please see lines 172 to 233.

Comment 14 | Page 9, line 173: “Analyses pooled outcome data…”. This sentence does not make sense. The analyses didn’t pool the data, the authors did. Second, it is stated that data was pooled from each study but the sentence before states that they were pooled by study design.

Response: In line with these suggestions from the reviewer, we have clarified this section. Please see line 192.

Comment 15 | I think sensitivity analyses should be performed for: 1. Relationship for haemoglobin with VO2peak vs VO2max, 2. Study design.

Response: Thank you for these suggested analyses. First, this review only included studies that assessed V̇O2max, and thus excluded studies that assessed V̇O2peak. In this framework, the requested sensitivity analyses for the relationship for hemoglobin with V̇O2peak vs V̇O2max is beyond the a priori scope of the review. Second, stratified/subgroup analyses between study design are included in the review. From our perspective, there were no decision nodes within this systematic review process that generated a need for a sensitivity analysis — the sequence of decisions for inclusion were clearly objective and non-contentious [1]. 

[1] Cochrane Handbook for Systematic Reviews of Interventions, Part 2: General methods for Cochrane reviews, Chapter 9: Analysing data and undertaking meta-analyses, Section 9.7 Sensitivity analyses.

Comment 16 | Page 10, line 176: Can you please use Higgin’s I^2 statistic to report the statistical heterogeneity. It does have limitations (as does Q and tau^2), but it is a measure that explains the relative statistical heterogeneity and is also easier for the reader to understand (especially those that do not understand meta-analysis).

Response: Done. As suggested by the reviewer, the Higgin’s I2 statistic has been added to report the relative statistical heterogeneity of meta-analyses within the review. 

Comment 17 | How did you account for small sample bias or publication bias? I would suggest Egger’s for small sample size and Vevea and Hedges Weight-Function Model for Publication Bias.

Response: In line with the suggestion by the reviewer, Egger’s test was used to assess potential publication bias in meta-analyses [1]. This has been added as a method (please see lines 179 to 180) and the results of these tests are presented in the text (please see lines 266 to 271).

[1] https://training.cochrane.org/resource/identifying-publication-bias-meta-analyses-continuous-outcomes

RESULTS

Comment 18 | All results should be described in past tense.

Response: Thank you for this suggestion. Generally, we agree and have assured that results are described in past tense. As an important caveat, there are several atemporal facts which, according to standardized grammatical rules, are presented using present tense. These atemporal factors are primarily associated with presentation of results in the tables and figures of the manuscript, which exist without relation to time. As an example, from lines 236 to 237— “The process of study selection and article categorization is represented…[in Fig 1]”

Comment 19 | Please use the PRISMA headings as mentioned above.

Response: Thank you for this suggestion. As described in response to Comment 1 above, PRISMA checklist has informed Level 1 and Level 2 Headings throughout the manuscript.

Comment 20 | If stratified analyses were performed (e.g., by sex: combined, male, female), please provide results that describe if there were significance (p-value) differences for each analysis (either from the initial analysis or between (among) stratified groups. This should be completed for all results in text and in tables.

Response: Done. As described on lines 293 to 295, exploratory analyses stratifying meta-regression models by biological sex found no sex differences in the relationship between hemoglobin mass and V̇O2max values. In addition to describing if there were significant differences for each analysis, we have provided adequate information for the interested reader to perform additional analyses or impute findings. 

Comment 21 | Page 16, line 314: Why was the study removed? Was it the ‘only’ one study? This could provide a rationale for the removal, but needs to be explained to the reader.

Response: The study referenced by the reviewer represents a study of a ‘clinical population’. This study was removed in response to the first round of revisions and concern about combining studies that included healthy populations with clinical populations. Indeed, it was ‘only’ one study removed as appropriately described in the text. Rationale for the removal has now been explicitly stated for the reader— please see lines 339 to 342.

DISCUSSION

Comment 24 | General: Please use or at least try to address PRISMA sub-headings. 1. General interpretation of results, 2. Limitations of evidence included in the review, 3. Limitations of review process, 4. Implications for practice, policy and future research.

Response: As requested, we have addressed the PRISMA sub-headings within the text, including:

1. General interpretation of results (paragraph 1 of discussion),

2. Limitations of evidence included in the review (limitations section of discussion),

3. Limitations of review process (limitations section of discussion),

4. Implications for practice, policy, and future research (ultimate paragraph of discussion entitled ‘conclusion’). 

Comment 25 | Page 24, line 497: What do you mean by simplistic design?

Response: Because we limited our focus to a single relationship between hemoglobin and V̇O2max, our designed approach did not consider the potential confounding effects of other relevant physiological factors, such as blood volume, plasma volume, maximum cardiac output, and muscle oxygen diffusivity. In this context, our primary analyses were simplistic — our analyses treated a complex biological relation as if it were much simple than it is. This section has been clarified—please see lines 525 to 530. 

Comment 26 | Page 24, line 499: What are you referring to by “more complex statistical methods”? You can use aggregate data to perform multivariate analyses. Limitations in the complexity of statistical analyses are probably due to the software that used to conduct the analyses. Software packages available in ‘R’ make it very easy to conduct multivariate analyses. The primary (and albeit significant) limitation using aggregate data is the fallacy of regression to the mean, which occurs when using continuous data of population characteristics to perform meta-regressions. However, this limitation occurs regardless of the type of analyses (i.e., univariate or multivariate analyses).

Comment 27 | Page 24, line 504: “For example…” This point is only a limitation if you choose not to run stratified or meta-regressions on intervention characteristics, which were completed for some variables. You can say that the limitations, specifically with your review was that you chose not to run these analyses for which there may be valid reasons which should be stated (e.g., the data was not available, etc.). But I would explain why you chose not examine the variables listed here. However, it is not a limitation to the approach as it is best to include all studies and then find out the source of statistical heterogeneity by means of the clinical heterogeneity.

Response: We have elected to respond to comments 26 and 27 from the reviewer – which both correspond to the same section of the manuscript—using one response. We agree with these sage remarks from the reviewer. We have removed the phrase ‘more complex statistical methods’, and more explicitly described the limitations of our approach. Please find the revised section on lines 524 to 550.

Comment 28 | Page 25, line 513: Good point. Though the primary differences in haemoglobin would be concentration versus mass. Please cite the literature for your examples. If you believe that your results show that there was no difference between the two measurement types, please explain this to the reader in further detail as it is an important finding that conflicts with current literature.

Response: Done. We have cited the literature for our examples, please find this citation both in the methods (line 129) and limitations section of the discussion (line 547). We do not believe that there is no difference between the two measurement types and our analyses did not evaluate the potential difference between the two measurement types. 

CONCLUSION

Comment 29 | Page 25, line 522: The results suggest that the pooled effect measure changed after addressing measures of clinical heterogeneity (though, it was not stated in the results section of the differences in modifiers were considered to be significant (no p-value to accompany the change in variance). Therefore, given the available results, I am not sure if this statement is true.

Response: Point well taken. We have removed the phrase “across a heterogeneous pool of studies”, and the concluding statement is now true without ambiguity. 

Comment 30 | Page 25, line 525: I would argue that there is still bias in the methods used to conduct the synthesis of the results. However, there are efforts within the manuscript to limit the degree of bias.

Response: We agree. The term “unbiased” has been removed and replaced with “comprehensive”. 

 

Reviewer 2

I have reviewed the document submitted by the authors again and have observed that the changes made in this second manuscript are minimal compared to the first, except for some statistical treatments (heterogeneity, Tau). The observations made in the first round are mainly included in a broad section of limitations that have been added to this second document.

The fundamental objections remain present in this second manuscript. Nevertheless, considering the magnitude of the effort made, the number of articles reviewed, and the fact that the limitations of the study are now explicitly stated based on the observations made, I believe that the article can be published. I think it is sufficiently clear what the real scope of the work's conclusions may be for non-expert readers in the field.

Response: Thank you for the review of the manuscript. We agree that the limitations section has provided additional and relevant context for the appropriate interpretation of these data.

---

## [Decision Letter · Decision Letter 2]

26 Sep 2023

PONE-D-22-29015R2The relationship between hemoglobin and V̇O2max: a systematic review and meta-analysisPLOS ONE

Dear Dr. Senefeld,

Thank you for submitting your manuscript to PLOS ONE. After careful consideration, we feel that it has merit but does not fully meet PLOS ONE’s publication criteria as it currently stands. Therefore, we invite you to submit a revised version of the manuscript that addresses the points raised during the review process.

I have asked a statistician to assist us because of the concerns raised by the reviewer. The comments would seem minor to me but feel free to address the best you can, in an efficient manner, the changes requested. If needed, you can also include supplementary data to save space in the main manuscript. Sorry for this but we hope you understand we all are working to publish the best possible article.

We look forward to receiving your revised manuscript.

Kind regards,

Daniel Boullosa

Academic Editor

PLOS ONE

Reviewers' comments:

Reviewer's Responses to Questions

**Comments to the Author**

1. If the authors have adequately addressed your comments raised in a previous round of review and you feel that this manuscript is now acceptable for publication, you may indicate that here to bypass the “Comments to the Author” section, enter your conflict of interest statement in the “Confidential to Editor” section, and submit your "Accept" recommendation.

Reviewer #1: (No Response)

Reviewer #3: (No Response)

2. Is the manuscript technically sound, and do the data support the conclusions?

Reviewer #1: Partly

Reviewer #3: Yes

3. Has the statistical analysis been performed appropriately and rigorously? 

Reviewer #1: No

Reviewer #3: Yes

4. Have the authors made all data underlying the findings in their manuscript fully available?

Reviewer #1: Yes

Reviewer #3: Yes

5. Is the manuscript presented in an intelligible fashion and written in standard English?

Reviewer #1: Yes

Reviewer #3: Yes

6. Review Comments to the Author

Reviewer #1: The authors have done a great job addressing the feedback that I provided. They have made significant improvements to the quality of their manuscript.

The only issue remaining is that all linear regressions must be removed from the analysis as they do not consider both within and between study variance. A linear regression can be used for individual patient data sets. However, it is not appropriate for aggregate data. If the measure of uncertainty (i.e., standard error) is unavailable then the data cannot not be pooled. The authors indicate that these analyses provide an important comparison to other work, which may be true. However, the methods used to conduct the analyses are incorrect.

Reviewer #3: The present is a high quality paper. Only minor issues

Abstract: it may be worth to have only one primary end point

methods: it should be added hiow many authors performed the search

methods: it should be added if random or fixed effect was used

methods; subgroup analysis according to kind of studies should be added

7. PLOS authors have the option to publish the peer review history of their article (what does this mean?). If published, this will include your full peer review and any attached files.

Reviewer #1: No

Reviewer #3: **Yes: **Fabrizio D'Ascenzo

---

## [Author Response · Author response to Decision Letter 2]

28 Sep 2023

Academic Editor

Thank you for submitting your manuscript to PLOS ONE. After careful consideration, we feel that it has merit but does not fully meet PLOS ONE’s publication criteria as it currently stands. Therefore, we invite you to submit a revised version of the manuscript that addresses the points raised during the review process.

I have asked a statistician to assist us because of the concerns raised by the reviewer. The comments would seem minor to me but feel free to address the best you can, in an efficient manner, the changes requested. If needed, you can also include supplementary data to save space in the main manuscript. Sorry for this but we hope you understand we all are working to publish the best possible article.

Response: As described previously, thank you for spearheading this review process, which has augmented the quality and potential impact of this manuscript. 

Reviewer 1

The authors have done a great job addressing the feedback that I provided. They have made significant improvements to the quality of their manuscript.

Response: We thank the reviewer for the thorough and constructive reviews of the manuscript. 

The only issue remaining is that all linear regressions must be removed from the analysis as they do not consider both within and between study variance. A linear regression can be used for individual patient data sets. However, it is not appropriate for aggregate data. If the measure of uncertainty (i.e., standard error) is unavailable then the data cannot not be pooled. The authors indicate that these analyses provide an important comparison to other work, which may be true. However, the methods used to conduct the analyses are incorrect.

Response: All linear regressions have been removed from the manuscript. 

Reviewer 3

The present is a high quality paper. Only minor issues

Response: Thank you for the review of the manuscript and supportive comment.

Abstract: it may be worth to have only one primary end point

Response: Indeed, we do have one primary end point. As described in the ‘objectives’ section of the manuscript (please see lines 74 to 79), the primary end point of this systematic review and meta-analysis aimed to evaluate the relationship between hemoglobin levels and V̇O2max. Because both hemoglobin levels and V̇O2max may be described using multiple metrics, our primary end point is associated with several different analyses. 

methods: it should be added hiow many authors performed the search

Response: This important information has been added—please see lines 92 to 93.

methods: it should be added if random or fixed effect was used

Response: We used random-effects models—please see lines 187 to 189.

methods; subgroup analysis according to kind of studies should be added

Response: Indeed, as a secondary objective, prespecified subgroup analyses were performed according to the kind of study. Separate analyses were included for observational studies and interventional studies, and additional analyses performed based on intervention (blood transfusion or donation, erythropoiesis stimulating agents, dietary supplementation, environmental hypoxia, or aerobic (de)training). We have revised the methods to more clearly delineate the subgroup analyses – please see lines 215 to 217.

---

## [Editor Report · Decision Letter 3]

2 Oct 2023

The relationship between hemoglobin and V̇O2max: a systematic review and meta-analysis

PONE-D-22-29015R3

Dear Dr. Senefeld,

We’re pleased to inform you that your manuscript has been judged scientifically suitable for publication and will be formally accepted for publication once it meets all outstanding technical requirements.

Kind regards,

Daniel Boullosa

Academic Editor

PLOS ONE
---

## [Editor Report · Acceptance letter]

4 Oct 2023

PONE-D-22-29015R3 

The relationship between hemoglobin and V̇O_2_max: a systematic review and meta-analysis 

Dear Dr. Senefeld:

I'm pleased to inform you that your manuscript has been deemed suitable for publication in PLOS ONE. Congratulations! Your manuscript is now with our production department. 

Kind regards, 

on behalf of

Dr. Daniel Boullosa 

Academic Editor

PLOS ONE